# On the Byzantine-Resilience of Distillation-Based Federated Learning

**Christophe Roux**[*]**, Max Zimmer**[*] **& Sebastian Pokutta**
Department for AI in Society, Science, and Technology, Zuse Institute Berlin, Germany
Institute of Mathematics, Technische Universität Berlin, Germany
`{roux,zimmer,pokutta}@zib.de`

## Abstract

Federated Learning (FL) algorithms using Knowledge Distillation (KD) have received increasing attention due to their favorable properties with respect to privacy, non-i.i.d. data and communication cost. These methods depart from transmitting model parameters and instead communicate information about a learning task by sharing predictions on a public dataset. In this work, we study the performance of such approaches in the byzantine setting, where a subset of the clients act in an adversarial manner aiming to disrupt the learning process. We show that KD-based FL algorithms are remarkably resilient and analyze how byzantine clients can influence the learning process. Based on these insights, we introduce two new byzantine attacks and demonstrate their ability to break existing byzantine-resilient methods. Additionally, we propose a novel defence method which enhances the byzantine resilience of KD-based FL algorithms. Finally, we provide a general framework to obfuscate attacks, making them significantly harder to detect, thereby improving their effectiveness.

## 1 Introduction

Federated Learning (FL) allows training machine learning models while keeping data private. The most common FL algorithm is Federated Averaging (FedAVG) (McMahan et al., 2017), where clients train models on their local data and share the updated model parameters with a central server. The server combines these parameters and sends back an updated model. While FedAVG has found success in diverse applications (Rieke et al., 2020; Ramaswamy et al., 2019; Yang et al., 2018; Sheller et al., 2019; Chen et al., 2019), it has notable limitations such as high communication costs due to repeatedly transmitting the model parameters, susceptibility to inversion attacks which reconstruct private data from the parameters (Melis et al., 2018; Nasr et al., 2019), or performance degradation due to non identically independently distributed (i.i.d.) local data among the clients (Kairouz et al., 2019). Most importantly for this work, standard FedAVG lacks resilience to byzantine clients, i.e., clients which behave adversarially. This has fueled efforts to create provably byzantine-resilient FedAVG variants (Allen-Zhu et al., 2021; Alistarh et al., 2018; Mhamdi et al., 2018; Chen et al., 2017; Zhu et al., 2023).

A recent line of work uses Knowledge Distillation (KD) in order address some of these challenges (Hinton et al., 2015). In general, KD transfers knowledge by training a *student* model on the aggregated predictions of multiple *teacher* models. For FL, it can enhance or replace the transmission of model parameters by sharing the clients' predictions on a public dataset. The server then uses KD to distill this information into its model. Such methods offer reduced communication overhead, enhanced privacy and robustness to non-i.i.d. data, but they are not well explored in the byzantine setting (Sattler et al., 2020; 2021; Lin et al., 2020; He et al., 2020; Cheng et al., 2021; Papernot et al., 2017; Chang et al., 2019; Gong et al., 2022; Fan et al., 2023).

In this work, we investigate KD-based FL algorithms in the byzantine setting, restricting ourselves to classification. While many different algorithms have been proposed in this area, we focus on a prototypical variant, referred to as Federated Learning using Distillation (FedDistill) because of

---

[*]Equal contribution

its simplicity and the fact the clients communicate with the server *only* through predictions on an unlabeled public dataset. We analyze how this communication modality–sharing predictions instead of model parameters–impacts the ability of byzantine clients to disrupt training, demonstrating that standard FedDistill offers greater byzantine-resilience than standard FedAVG.

We identify two key factors underlying this resilience: First, byzantine clients in FedDistill can only modify predictions, which are constrained to the bounded and relatively low-dimensional probability simplex. This contrasts sharply with FedAVG, where attacks occur in the unbounded, high-dimensional parameter space. Second, byzantine clients in FedDistill can only influence the server model indirectly through the distillation objective, whereas in FedAVG they directly manipulate the model parameters.

Guided by these insights, we introduce two novel FedDistill-specific attacks that successfully bypass existing byzantine-resilient FedDistill variants. In response, we propose a new defence mechanism that detects and filters byzantine clients by leveraging client information from both past and current communication rounds. Additionally, we present a general framework to obfuscate attacks, making byzantine clients harder to detect and hence calling for further research on defence mechanisms. To summarize, our contributions are as follows:

1. We analyze FedDistill, a prototypical KD-based FL algorithm, in the byzantine setting, focusing on its unique communication method of transmitting predictions rather than parameters to the server. Our analysis reveals that byzantine clients have only a limited impact on vanilla FedDistill, rendering it more resilient than vanilla FedAVG.

2. We introduce LMA and CPA, two novel and effective byzantine attacks which are specifically designed for FedDistill. Our extensive experiments demonstrate stronger disruption of the training process than previous approaches, even when comparing against existing byzantine-resilient FedDistill variants.

3. In response, we propose ExpGuard, a novel defence mechanism for FedDistill that incorporates information on client behavior in both current and past communication rounds. ExpGuard significantly improves byzantine resilience compared to previous methods that rely solely on robust aggregation applied independently to each sample.

4. Finally, we propose HIPS, a general method to obfuscate byzantine attacks, making them harder to detect. In the presence of defence mechanisms, HIPS improves the effectiveness of byzantine attacks.

Our findings represent an important step forward in the analysis of byzantine FL, providing insights into both the vulnerabilities and defences of KD-based FL systems. By developing novel attack and defence mechanisms, we further advance the robustness of KD-based FL.

**Outline.** We begin by reviewing prior work on byzantine FL and applications of KD in FL in Section 2. We then introduce our problem setting, detail the FedDistill algorithm, and describe our experimental methodology in Section 3. In Section 4, we analyze how byzantine clients affect the server model in both FedDistill and FedAVG, demonstrating that FedDistill is more resilient by design. We present our two novel byzantine attacks in Section 5.1, followed by our defence mechanism in Section 5.2. Finally, we describe HIPS in Section 5.3, our technique for making byzantine attacks harder to detect by obfuscating malicious behavior.

## 2 RELATED WORK

**KD in FL.** One common approach of employing KD in FL is to transmit information from the clients to the server via predictions on a publicly available unlabeled dataset (Li & Wang, 2019; Chang et al., 2019; Cheng et al., 2021; Gong et al., 2022; Lin et al., 2020; Sturluson et al., 2021; Bistritz et al., 2020). Some works (Li & Wang, 2019; Cheng et al., 2021) assume the availability of a *labeled* public dataset. Lin et al. (2020) and Sturluson et al. (2021) study augmenting FedAVG-type algorithms using KD. In this setting clients send model parameters as well as predictions on a public dataset to the server, meaning that they can disturb the training process both via the parameters and predictions.

**Byzantine FedAVG.** Some byzantine-resilient variants replace the mean with more robust alternatives for aggregating client updates (Blanchard et al., 2017; Mhamdi et al., 2018; Yin et al., 2021; Chen et al., 2017). Additionally, Karimireddy et al. (2022); Allouah et al. (2023) propose simple preprocessing steps that can be used to improve the byzantine-resilience. These approaches are orthogonal to the defences we propose in this paper and could be used to further improve their performance. However, replacing the mean with robust aggregation methods may disregard the honest clients' predictions, which can hinder performance, and remain vulnerable to more sophisticated attacks (Xie et al., 2020; Baruch et al., 2019; Shejwalkar & Houmansadr, 2021). These issues have been addressed by considering historical client behavior throughout the training process (Alistarh et al., 2018; Allen-Zhu et al., 2021; Karimireddy et al., 2021). The previously mentioned approaches rely on the assumption that the data held by the clients are homogeneous (i.i.d.). As the data becomes more heterogeneous (non-i.i.d.), it becomes harder to distinguish between a malicious update and an update created by dissimilar data which holds valuable information. Some recent works introduced variants of FedAVG that have limited byzantine-resilience in the heterogeneous setting (Karimireddy et al., 2022; El-Mhamdi et al., 2021). Note however that even in the benign setting, non-i.i.d. data poses serious issues, e.g. (Kairouz et al., 2019), which is why we restrict ourselves to the i.i.d. setting in the main paper. However, we show that our attacks and defences are also effective in the non-i.i.d. setting in Appendix D.

**Byzantine FedDistill.** Federated Robust Adaptive Distillation (Sturluson et al., 2021) is a variant of FedAVG that uses KD to increase its byzantine-resilience in the non-i.i.d. setting. After local training, the server aggregates the updated parameters of the clients using a weighted sum. The weights are computed based on how much the prediction of the clients conform to the prediction of the majority. Since this method relies on sharing the parameters and predictions on a public dataset, attacks as well as defences operate both in the prediction and parameter space. In contrast, the goal of this work is to isolate the effects of attacks and defences that operate purely in the prediction space. Most relevant to our work, Federated Ensemble Distillation (Chang et al., 2019) uses predictions on a public dataset for both client-server and server-client communication. The server only aggregates client predictions, sends them back, and clients train both the public dataset using the aggregated predictions as labels and their local data. They introduce Cronus, a byzantine-resilient aggregation method, which improves the resilience but suffers from bad empirical performance in the benign setting. Both of these works concentrate on designing byzantine-resilient algorithms and give limited attention to finding more effective byzantine attacks, testing their algorithms only against simple label flip heuristics.

## 3 PRELIMINARIES

We limit ourselves to the supervised classification setting and assume that there are $N$ clients in total. Let $\mathcal{H}$ be the set of *honest* clients and $\mathcal{B}$ the set of *byzantine* clients, assuming that an $\alpha$-fraction of the clients are byzantine where $\alpha < 0.5$, which is a standard assumption in byzantine FL to expect reasonable outcomes. Each honest client $i \in [N]$ holds a *private* dataset $\mathcal{D}_i$ sampled from the same distribution as the publicly available but *unlabeled* dataset $\mathcal{D}_{\text{pub}}$. The objective of FedDistill is to train a classifier based only on $\mathcal{D}_{\text{pub}}$ using the predictions of the clients, which are computed and transmitted in multiple communication rounds.

Given a sample $x$, we denote by $Y_i^t(x)$ the prediction of client $i$ in communication round $t$ using neural network parameters $w_i^t$. The honest clients infer their predictions by using their local model $h(x, w_i^t)$ while the byzantine clients can send *any* vector in $\Delta_c$, the $c$-dimensional probability simplex with $c$ being the number of classes. We define $Y_i^t(\mathcal{D}_{\text{pub}}) \stackrel{\text{def}}{=} \{Y_i^t(x) : x \in \mathcal{D}_{\text{pub}}\}$ as the set of predictions of client $i$ on every datapoint in $\mathcal{D}_{\text{pub}}$. We sometimes drop the dependence on $x$ for simplicity. Further, we denote the aggregated prediction of all clients by $\bar{Y}^t(x)$ and the aggregated parameters of all clients by $\bar{w}_i$. At the end of each a communication round $t$, the server has received the *individual* client predictions $Y_i^t(\mathcal{D}_{\text{pub}})$ of each client $i$, but is unaware which clients are byzantine.

We assume that the byzantine clients have access to the *individual* predictions of the honest clients on $\mathcal{D}_{\text{pub}}$ before sending their own predictions, allowing to base their attack on them. Byzantine clients are permitted to collude. However, their identities remain consistent across iterations, ensuring that the server can discern which client generated which set of predictions. These assumptions are analogous to byzantine FedAVG, where the byzantine clients also cannot change identity, can collude

and are assumed to have access to the individual parameter updates sent to the server by the clients (Allen-Zhu et al., 2021; Alistarh et al., 2018; Karimireddy et al., 2021).

**FedDistill vs. FedAVG comparison.** Algorithm 1 showcases FedDistill and FedAVG in a unified way. At the beginning of each communication round, the server broadcasts its parameters $\bar{w}_i$ to the clients (Line 2). The clients train on their local datasets starting from these parameters (Line 3). In FedAVG, clients directly send their updated parameters $w_i^{t+1}$ to the server (Line 4) and the server aggregates these parameters to obtain the new server parameters $\bar{w}_i^{t+1}$ (Line 5). In FedDistill, clients transmit information about the learning task via their predictions on a public dataset (Line 6) and the server aggregates these and updates its model

---

**Algorithm 1** Federated Learning (FL)

---

1: **for** communication round $t = 0$ to $T - 1$ **do**
2:     SERVER: Broadcast parameters $\bar{w}_i$ to the clients
3:     CLIENTS: Train on private datasets

    ___ **FedAVG** ___________________________

4:     CLIENTS: Send updated parameters to server
5:     SERVER: Aggregate parameters to obtain $\bar{w}_i^{t+1}$

    ___ **FedDistill** ________________________

6:     CLIENTS: Send public dataset predictions to server
7:     SERVER: Train on public dataset with aggregated client predictions to obtain $\bar{w}_i^{t+1}$

---

8: **end for**
9: **Output:** $\bar{w}_T$

---

by training on the public dataset using the aggregated prediction as labels (Line 7). Since in FedDistill, the server can only be influenced by the clients predictions on $\mathcal{D}_{\text{pub}}$, we can study the byzantine impact given this communication modality in isolation.

**Experimental setup.** We evaluate FedDistill on the CIFAR-10/100 (Krizhevsky et al., 2009), CINIC-10 (Darlow et al., 2018), and Clothing1M (Xiao et al., 2015) datasets using the ResNet (He et al., 2016) and WideResNet (Zagoruyko & Komodakis, 2016) architectures. We keep 5% of the training datasets for validation and split the remaining data evenly among the clients. Each experiment is performed with multiple random seeds and we report mean and standard deviation.

To ensure a feasible computational study, we set several parameters of the algorithms to fixed values. Specifically, we set the number of clients to 20 and the number of communication rounds to 10. We conduct ablations on these parameters in Appendix D. We set the number of total local epochs each client performs, i.e., training on their private datasets, to a fixed value depending on the dataset used. Similarly, we set a total number of communications and uniformly distribute these communications among the local epochs. We train clients and server using SGD with weight decay and a linearly decaying learning rate from 0.1 to 0, momentum is set to 0.9. Appendix B contains a detailed account of the experimental setup. Our code is available at github.com/ZIB-IOL/FedDistill.

## 4 THE BYZANTINE SETTING

Due to the different communication modalities of FedAVG and FedDistill, byzantine clients influence the training in different ways. We refer to these modes of influence as *threat vectors*. In this section, we investigate these threat vectors and their impact, first comparing the byzantine-resilience of vanilla FedAVG and FedDistill as a motivating example.

Figure 1 compares how FedAVG and FedDistill are impacted by byzantine clients when using two naive attacks. We measured the final test accuracy when varying the fraction of byzantine clients $\alpha$. *FedAVG: GN* denotes FedAVG where byzantine clients send Gaussian noise instead of the updated model parameters. Analogously, *FedDistill: RLF* refers to FedDistill where for each sample, all byzantine clients return the same random label. While FedAVG collapses with only 10% byzantine clients ($\alpha = 0.1$), FedDistill is remarkably byzantine-resilient, even at $\alpha = 0.45$. These experiments underscore the inherent robustness of FedDistill, which we now discuss from a theoretical perspective. Note that Figure 1 serves as a motivating example, more advanced aggregation schemes for FedAVG could easily counter the Gaussian noise attack. See Appendix D for a comparison between FedDistill and state of the art byzantine-resilient FedAVG methods.

**Threat vectors in FedAVG and FedDistill.** In standard FedAVG, the updated parameters are aggregated by taking their mean. In this case, we can highlight the threat vector by decomposing the aggregation step (Line 5) into an honest and a byzantine part as follows,

$$\bar{w}_i^{t+1} \leftarrow \frac{1}{N} \sum_{i \in \mathcal{H}} w_i^{t+1} + \underbrace{\frac{1}{N} \sum_{i \in \mathcal{B}} w_i^{t+1}}_{\text{FedAVG threat vector}} .$$

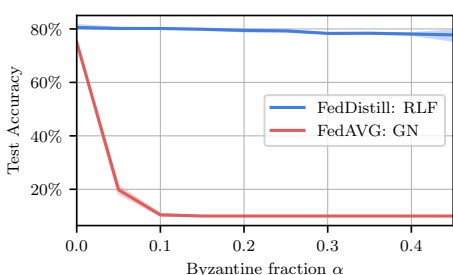

The byzantine clients can directly influence the new server parameters via their parameters $w_i^{t+1}$. Setting the weights of a single client to $\tilde{w} N - \sum_{i \neq j} w_i^{t+1}$ suffices to achieve $\bar{w}_i^{t+1} = \tilde{w}$, where $\tilde{w}$ can be chosen freely. In other words, just one byzantine client is sufficient to arbitrarily perturb the new server parameters.

Figure 1: ResNet-18 on CINIC-10: Final test accuracy of FedAVG and FedDistill, varying the fraction of byzantine clients for two naive attacks.

On the other hand, in FedDistill the clients can only alter the server model via their predictions on the public dataset $\mathcal{D}_{\text{pub}}$. The server distills the knowledge from the clients based on their predictions on the public dataset (Line 7). Recall that for a sample $x \in \mathcal{D}_{\text{pub}}$, the mean of the clients' predictions in round $t$ is denoted by $\bar{Y}_i^{t+1}(x) \stackrel{\text{def}}{=} \frac{1}{N} \sum_{i=1}^{N} Y_i^{t+1}(x)$ and the server solves the optimization problem

$$\min_w \sum_{x \in \mathcal{D}_{\text{pub}}} \mathcal{L}(h(x, w), \underbrace{\bar{Y}_i^{t+1}(x)}_{\text{FedDistill threat vector}}), \tag{$\mathcal{P}_{\text{distill}}$}$$

where $\mathcal{L}$ is a loss function such as the Cross-Entropy Loss (CEL) or Mean-Squared Error (MSE). By decomposing $\bar{Y}_i^{t+1}(x)$ into a byzantine $\bar{Y}_{\mathcal{B}}^{t+1}(x) \stackrel{\text{def}}{=} \frac{1}{\alpha N} \sum_{i \in \mathcal{B}} Y_i^{t+1}(x)$ and an honest part $\bar{Y}_{\mathcal{H}}(x) \stackrel{\text{def}}{=} \frac{1}{(1-\alpha)N} \sum_{i \in \mathcal{H}} Y_i^{t+1}(x)$, we can highlight the threat vector more specifically,

$$\bar{Y}_i^{t+1}(x) = \underbrace{\alpha \bar{Y}_{\mathcal{B}}^{t+1}(x)}_{\text{FedDistill threat vector}} + (1 - \alpha) \bar{Y}_{\mathcal{H}}^{t+1}(x), \tag{1}$$

where $\alpha$ is the fraction of byzantine clients.

**The byzantine resilience of FedDistill.** There are two main reasons why vanilla FedDistill is less susceptible to byzantine attacks than vanilla FedAVG. First, the influence of byzantine clients on the server parameters in FedDistill is indirect. They can only affect the distillation objective ($\mathcal{P}_{\text{distill}}$) through their predictions, which must then influences the server's optimization procedure. This contrasts with FedAVG, where byzantine clients can directly manipulate the server parameters through their parameter updates. Second, the threat vector in FedDistill is confined to the bounded probability simplex $\Delta_c$, limiting the impact of perturbations, unlike in FedAVG, where the threat vector operates in the higher-dimensional, unbounded parameter space.

Since the server cannot distinguish between honest and byzantine clients, it has to resort to distilling the knowledge from the aggregated predictions from *all* clients by minimizing ($\mathcal{P}_{\text{distill}}$), despite its true objective being to distill knowledge only from the *honest* clients by solving

$$\min_w \sum_{x \in \mathcal{D}_{\text{pub}}} \mathcal{L}(h(x, w), \bar{Y}_{\mathcal{H}}^{t+1}(x)). \tag{$\mathcal{P}_{\text{honest}}$}$$

We show that if a parameter $w$ is a stationary point of the perturbed objective ($\mathcal{P}_{\text{distill}}$), it also lies within a neighborhood of a stationary point of the true objective ($\mathcal{P}_{\text{honest}}$). Therefore, running stochastic gradient descent on ($\mathcal{P}_{\text{distill}}$) still enables the server to converge to a neighborhood of the solution it would reach with ($\mathcal{P}_{\text{honest}}$). The key intuition behind this result is that ($\mathcal{P}_{\text{distill}}$) and ($\mathcal{P}_{\text{honest}}$) differ only in their target labels, allowing us to demonstrate that the difference between their computed gradients scales linearly with the distance between $\bar{Y}_{\mathcal{H}}$ and $\bar{Y}_{\mathcal{B}}$:

$$\left\| \nabla_w \mathcal{L}(w, \bar{Y}_{\mathcal{H}}^{t+1}) - \nabla_w \mathcal{L}(w, \bar{Y}_i^{t+1}) \right\| \leq C\alpha \left\| \bar{Y}_{\mathcal{H}}^{t+1} - \bar{Y}_{\mathcal{B}}^{t+1} \right\| \leq \sqrt{2} C\alpha,$$

where we used the shorthand $\mathcal{L}(w, Y) \stackrel{\text{def}}{=} \mathcal{L}(h(w, x), Y(x))$. Here, $C$ is a constant independent of the prediction and depending only on the parameter $w$ and the data $x \in \mathcal{D}_{\text{pub}}$. Note that the error grows proportionally to the fraction of byzantine clients $\alpha$ and the distance between the aggregated honest and the byzantine predictions. Furthermore, since both predictions lie in $\Delta_c$, their distance is inherently bounded by $\sqrt{2}$. See Appendix E.1 for the full statement and proof of the following informal theorem.

**Theorem 1** (Informal). *If $\tilde{w}$ is a stationary point of $(\mathcal{P}_{\text{distill}})$, then it is also an $\mathcal{O}(C^2\alpha^2)$-approximate stationary point of $(\mathcal{P}_{\text{honest}})$, where $C > 0$ is a constant independent of the client predictions. Further, in expectation, running Stochastic Gradient Descent (SGD) on $(\mathcal{P}_{\text{distill}})$ to achieve an $\varepsilon$-approximate stationary point yields an $\mathcal{O}(\varepsilon + C^2\alpha^2)$-approximate stationary point of $(\mathcal{P}_{\text{honest}})$.*

## 5 ATTACKING AND DEFENDING KD-BASED FL

In the previous section, we compared standard variants of FedDistill and FedAVG in terms of their susceptibility to byzantine attacks. Building on our insights into the threat vectors, this section focuses on developing strategies to attack and defend KD-based FL. We first introduce two effective attacks in Section 5.1, specifically tailored to exploit the threat vector of FedDistill. To counter these attacks, we propose a novel defence mechanism in Section 5.2, which significantly enhances byzantine-resilience compared to prior methods. Finally, Section 5.3 presents a general framework for obfuscating byzantine attacks on KD-based FL, making them much harder to detect. In the following, we omit the time superscripts on the prediction to simplify the notation.

### 5.1 ATTACKING FEDDISTILL: LMA & CPA

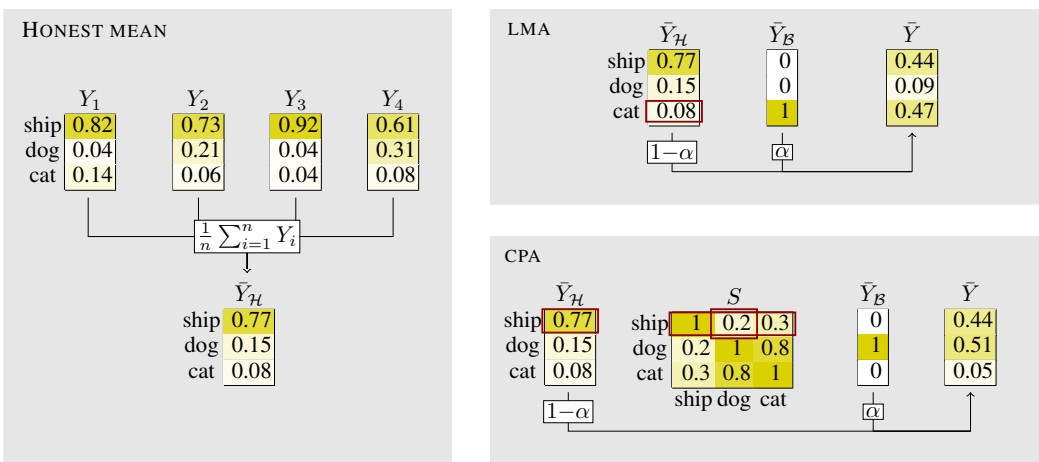

Figure 2: Attack procedures for a three-class classification problem with four honest and three byzantine clients, i.e., $\alpha = 3/7$. The left part of the figure shows the computation of the honest mean. LMA (upper right) assigns probability one to the least likely class based on the honest mean $\bar{Y}_{\mathcal{H}}$ and CPA (lower right) assigns probability one to the class that is least similar to the most likely class of $\bar{Y}_{\mathcal{H}}$, according to the similarity matrix $S$. Note that all computations are done per sample $x$, which was omitted from the notation for legibility.

In FedDistill, byzantine clients can only degrade the server model through their predictions. Since the threat vector is confined to the $c$-dimensional probability simplex $\Delta_c$ rather than the parameter space, many attacks from the byzantine FedAVG literature, such as gradient sign flipping, cannot be applied to FedDistill. Previous work on byzantine FedDistill has been limited to basic label-flipping attacks.

While the server trains on the aggregated predictions from *all* clients $\bar{Y}(\mathcal{D}_{\text{pub}})$, its true objective is to match the mean predictions $\bar{Y}_{\mathcal{H}}(\mathcal{D}_{\text{pub}})$ made by the *honest* clients. Hence, one can interpret the aim of the byzantines as selecting their predictions $\bar{Y}_{\mathcal{B}}(\mathcal{D}_{\text{pub}})$ to maximize the disparity between the aggregated predictions $\bar{Y}(\mathcal{D}_{\text{pub}})$ and the predictions of the honest clients $\bar{Y}_{\mathcal{H}}(\mathcal{D}_{\text{pub}})$.

**Loss Maximization Attack (LMA).** One way to measure the difference between $\bar{Y}(\mathcal{D}_{\text{pub}})$ and $\bar{Y}_{\mathcal{H}}(\mathcal{D}_{\text{pub}})$ is via the loss function $\mathcal{L}$ that the server aims to minimize. We propose LMA, which chooses byzantine predictions that *maximize* this loss for each sample $x \in \mathcal{D}_{\text{pub}}$. Specifically, for each sample, LMA solves the optimization problem

$$\max_{\bar{Y}_{\mathcal{B}}(x) \in \Delta_c} \mathcal{L}\left((1-\alpha)\bar{Y}_{\mathcal{H}}(x) + \alpha\bar{Y}_{\mathcal{B}}(x), \bar{Y}_{\mathcal{H}}(x)\right). \tag{2}$$

For typical loss functions, LMA corresponds to choosing the least likely class according to the aggregated predictions of all *honest* clients $\bar{Y}_{\mathcal{H}}(x)$, see Lemma 5 in Appendix E.2. For each $x \in \mathcal{D}_{\text{pub}}$, the byzantine clients compute the $\arg\min$ of the mean honest prediction $\bar{Y}_{\mathcal{H}}(x)$, corresponding to the least likely class assigned to $x$, and then assign probability 1 to this class, i.e., the byzantine prediction becomes $\mathbb{1}_j$, where $j \in \arg\min \bar{Y}_{\mathcal{H}}(x)$. An example is shown in the top right of Figure 2. According to $\bar{Y}_{\mathcal{H}}$, the least likely class is *cat*, hence all byzantine clients will assign probability one to *cat*. As a result, the mean of all clients $\bar{Y}$ predicts the class *cat*.

**Class Prior Attack (CPA).** Another way to measure how different two predictions are is to take information about the relationships between classes into account. For example, the class *dog* is more similar to the class *cat* than to the class *ship* . The second attack we propose, CPA, uses information about such inter-class relationships to select the most misleading predictions. To measure the similarity between the $c$ classes, CPA requires access to a similarity matrix $S \in \mathbb{R}^{c \times c}$. We obtain this similarity matrix by computing the sample covariance matrix of the predictions $Y(\mathcal{D}_{\text{pub}})$ of a pretrained model, i.e.,

$$S = \frac{1}{|\mathcal{D}_{\text{pub}}|} \sum_{x \in \mathcal{D}_{\text{pub}}} (Y(x) - \bar{Y})(Y(x) - \bar{Y})^T, \quad \bar{Y} \leftarrow \frac{1}{|\mathcal{D}_{\text{pub}}|} \sum_{x \in \mathcal{D}_{\text{pub}}} Y(x).$$

Then, for each sample $x$, CPA computes the class which is least similar to the class predicted by the honest clients per $\bar{Y}_{\mathcal{H}}(x)$, where similarity is measured using the similarity matrix $S$. The byzantine clients now assign probability 1 to that class, i.e., when $i = \arg\min \bar{Y}_{\mathcal{H}}(x)$, the byzantine clients predict $\mathbb{1}_j$ where $j \leftarrow \arg\min_j S_{ij}$ (see Figure 2). An example is shown in the bottom right of Figure 2. According to $\bar{Y}_{\mathcal{H}}$, the most likely class is *ship*, and based on $S$, *dog* is the least similar class to *ship*. Hence all byzantine clients will assign probability one to *dog*. As a result, the mean of all clients $\bar{Y}$ predicts the class *dog*.

Although using a pretrained model is not realistic in practice, we employ it to better understand CPA's effectiveness and potential capabilities. In a real-world setting, attackers could instead use the server model from previous rounds or leverage their prior knowledge about class relationships to construct the similarity matrix.

Figure 3 illustrates the effects of LMA and CPA over multiple communication rounds, where we compare to the naive baseline Random Label Flip (RLF) attack, where all byzantine clients assign the same random label to each sample. Both CPA and LMA are significantly more effective than RLF and heavily impact the final test accuracy. Notably, while the final test accuracy of both methods is similar, CPA has a stronger effect in the early rounds.

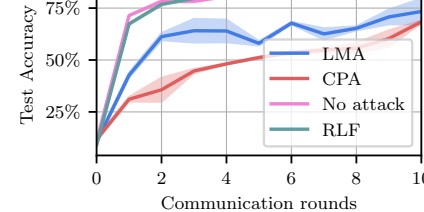

## 5.2 Defending FedDistill: ExpGuard

Unlike the baseline attack RLF, both LMA and CPA can severely disturb the training process. In order to improve the resilience against such attacks, we investigate replacing the mean aggregation of the clients' predictions with more robust aggregation methods.

Figure 3: ResNet-18 on CIFAR-10: Test accuracy evolution over communication rounds when attacking FedDistill with 9 byzantine out of overall 20 clients.

Let us first consider existing defence approaches, namely Cronus (Chang et al., 2019), a method based on filtering-based approaches from the high-dimensional robust statistics literature (Diakonikolas et al., 2017) and the geometric median (GM), a multivariate generalization of the median defined

as $\mathrm{GM}(Y_1, \ldots, Y_N) \stackrel{\text{def}}{=} \arg\min_{y \in \Delta_c} \sum_{i=1}^N \|Y_i - y\|_2$. Some commonly used robust aggregation methods further include the coordinate-wise trimmed mean or the coordinate-wise median; however, these cannot be used to aggregate predictions, as the aggregated vectors do not necessarily reside in the probability simplex, see Appendix E.3 for more details.

Filtering-based approaches compute the empirical covariance matrix of the data, obtain the leading eigenvector of that matrix and then project the data onto the subspace spanned by the eigenvector. The intuition behind these approaches is that if an adversary wants to perturb the mean by modifying a fraction of the data without creating obvious outliers, it has to set the datapoints relatively close to the benign data but all biased in a similar direction, such that the perturbation caused by the individual datapoints do not counteract each other. This causes the variance along this direction to increase.

In the setting discussed in Diakonikolas et al. (2017), outliers along this eigenvector are then filtered based on a specific rule relying on statistical assumptions. Since these assumptions do not hold in the FedDistill setting, Cronus relies on a simple heuristic: it filters out the 25% worst outliers, then recomputes the empirical covariance matrix and leading eigenvector based on the remaining 75% of the data and then filters out another 25% of the predictions, such that 50% of the original predictions remain.

Neither Cronus nor GM can be computed in closed-form and require numerical algorithms. Applying them to aggregated parameters in Byzantine FedAVG can lead to a large computational overhead. However, for FedDistill, we only aggregate predictions, which are of much smaller dimensions, making their computational requirements negligible compared to training (cf. Table 7, Appendix D).

**ExpGuard.** Both GM and Cronus only use information from the current datapoint $x \in \mathcal{D}_{\text{pub}}$. To further improve the resilience of FedDistill, we propose ExpGuard, which incorporates additional information available to the server from the predictions on other samples *within* each communication round as well as information about the predictions in *past* communication rounds.

ExpGuard, outlined in Algorithm 2, is a meta-algorithm that can be used to enhance any robust aggregation method. It works as follows: a robust aggregation method AGG is used to compute an outlier score $\sigma_i$ for each client based on its prediction on the whole dataset $\mathcal{D}_{\text{pub}}$. Then, these outlier scores are used to update the weight $p_i^t$ for each client based on the exponential weights algorithm (Littlestone & Warmuth, 1994) such that clients with low outlier scores $\sigma_i$ are assigned higher weight and the weight of clients with high outlier scores $\sigma_i$ is reduced.

---

**Algorithm 2** ExpGuard

1: **Input:** Predictions $Y_i^{t+1}(\mathcal{D}_{\text{pub}})$ and weights $p_i^t$ for all $i \in N$, aggregation method AGG.

2: Compute outlier scores:
$\sigma_i \leftarrow \mathrm{AGG}(Y_i^{t+1}(\mathcal{D}_{\text{pub}})), \forall i \in [n]$
3: Update weights:
$p_i^{t+1} \leftarrow p_i^t \exp(-\sigma_i), \forall i \in [n]$
4: Compute weighted sum for all $x \in \mathcal{D}_{\text{pub}}$:
$\bar{Y}_i^{t+1}(x) \leftarrow \frac{1}{\sum_{j=1}^n p_j^{t+1}} \sum_{i=1}^N p_i^{t+1} Y_i^{t+1}(x)$

5: **Output:** $\bar{Y}_i^{t+1}(\mathcal{D}_{\text{pub}}), p_i^{t+1}, \forall i \in [N]$

---

For any robust aggregation method, the outlier score can be computed as the sum of the distances between the clients prediction and the robust estimate for each $x \in \mathcal{D}_{\text{pub}}$. However, for filtering-based methods, we can skip the filtering, i.e., deciding which samples should be considered outliers, and simply use the distance of the clients prediction along the eigenvector as the outlier score. We refer to this approach as ExpGuard+Filter (EG+F). While ExpGuard substantially improves the byzantine resilience of all aggregation methods we tested, EG+F leads to the best results, see Table 8 in Appendix D. Therefore, we only show the results for EG+F in Table 1.

Table 1 presents a comparison of robust aggregation methods against our proposed attacks and the Random Label Flip baseline. The results show that robust aggregation methods improve robustness over simple mean aggregation and EG+F consistently outperforms both GM and Cronus across all datasets, highlighting the benefits of utilizing all available information. Interestingly, GM generally outperforms Cronus while ExpGuard+GM is less byzantine-resilient than EG+F, see Table 8 in Appendix D. This indicates that the poor performance of Cronus might be due to the filtering heuristic employed.

Table 1: FedDistill: 20 clients of which nine are byzantine ($\alpha = 0.45$). Final test accuracy averaged over multiple runs with standard deviation for different attacks and defences. BA refers to the baseline accuracy, i.e., the final accuracy of FedDistill if all clients are honest.

| | CINIC-10 (ResNet-18), BA=80.2±0.1 | | | | CIFAR-10 (ResNet-18), BA=87.7±1.2 | | | |
|---|---|---|---|---|---|---|---|---|
| | Mean | GM | Cronus | EG+F | Mean | GM | Cronus | EG+F |
| RLF | 76.9±0.4 | 79.3±0.3 | 76.6±0.0 | 78.8±0.1 | 84.6±0.1 | 85.4±0.6 | 84.7±0.3 | 85.4±0.0 |
| LMA | 54.6±1.2 | 75.0±0.6 | 71.1±1.7 | 77.4±1.1 | 73.4±8.6 | 83.3±0.2 | 80.6±2.3 | 85.4±0.1 |
| CPA | 45.9±0.4 | 71.2±5.2 | 65.9±3.3 | 79.2±0.2 | 68.4±0.8 | 78.4±0.9 | 74.5±0.6 | 85.5±0.8 |

| | CIFAR-100 (WideResNet-28), BA=66.8±0.5 | | | | Clothing1M (ResNet-50), BA=69.0±0.3 | | | |
|---|---|---|---|---|---|---|---|---|
| | Mean | GM | Cronus | EG+F | Mean | GM | Cronus | EG+F |
| RLF | 65.2±0.7 | 65.2±0.3 | 44.3±1.7 | 63.9±0.6 | 69.4±1.2 | 68.7±0.8 | 68.6±0.4 | 68.7±1.1 |
| LMA | 41.8±4.4 | 51.3±0.1 | 44.6±0.2 | 57.2±1.2 | 40.3±3.3 | 58.3±0.7 | 61.4±0.5 | 68.3±0.7 |
| CPA | 43.3±1.2 | 56.7±0.9 | 55.3±0.3 | 62.1±1.4 | 33.7±2.7 | 58.4±0.3 | 43.9±12.9 | 68.5±1.0 |

## 5.3 HIPS

The reason ExpGuard is so effective at mitigating the attacks in our experiments is that these attacks aim to maximally disrupt the learning process without considering detectability. However, in the presence of defences, a tradeoff arises between an attack's strength and its detectability. Even the most potent attack is ineffective if consistently detected and filtered out by a defence mechanism.

To address this tradeoff, we propose Hiding In Plain Sight (Hiding In Plain Sight (HIPS)), a method that obfuscates attacks to make them harder to detect. The key idea behind HIPS is that small perturbations to proper predictions can sufficiently disrupt the learning process while avoiding detection, as they are less distinguishable from honest predictions. We note that similar concepts have been explored in the byzantine FedAVG literature (Baruch et al., 2019; Xie et al., 2020), but as these approaches where designed for the parameter space, they do not apply to FedDistill.

For each sample $x$, HIPS operates by reducing the attack space from $\Delta_c$ to the convex hull of the honest predictions $\mathcal{A}(x) \overset{\text{def}}{=} \text{conv}\{Y_i(x) : i \in \mathcal{H}\}$ such that the predictions of the byzantine clients do not differ much from the honest predictions. We illustrate this concept in Figure 4 for a three-class classification problem. In Figure 4a, the byzantine clients can choose any point in $\Delta_3$, whereas in Figure 4b, due to the constraints imposed by HIPS, the attack space is reduced to $\mathcal{A}$. Given that HIPS introduces additional constraints to the attack space, it becomes necessary to modify the attacks accordingly. In the following, we omit the dependence on $x$ for legibility. We begin by extending CPA with HIPS.

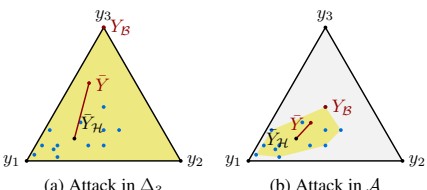

(a) Attack in $\Delta_3$     (b) Attack in $\mathcal{A}$

Figure 4: Attack spaces in $\Delta_3$: The blue dots represent the predictions by the honest clients and $\bar{Y}_{\mathcal{H}}$ is their mean. The attack space is highlighted. $\bar{Y}_{\mathcal{B}}$ is the byzantine prediction, and $\bar{Y}$ denotes the mean of *all* clients for $\alpha = 0.5$. The red line joining them represents the mean $\bar{Y}$ corresponding to different $\alpha \in [0, 0.5]$.

**Extending CPA with HIPS.** Recall that for each sample $x$, CPA computes the class least similar to the class predicted by the honest clients, measuring similarity using the similarity matrix $S$. Let $S_i$ denote the $i$-th row of $S$, i.e., $S_i$ is a vector containing the similarity of $i$ to all other classes. If $i$ is the class predicted according to the mean honest prediction $\bar{Y}_{\mathcal{H}}(x)$, then CPA can be rephrased as the solution to $\min_{y \in \Delta_c} y^T S_i$. Applying HIPS now requires restricting the constraints from $\Delta_c$ to $\mathcal{A}$.

Unlike CPA, this attack can no longer be computed in closed form. However, since $\min_{y \in \mathcal{A}} y^T S_i$ is a linear program, it is sufficient to consider the extreme points of $\mathcal{A}$, which correspond to the predictions of each honest client. Thus, the solution can still be efficiently obtained by evaluating the

function for each honest client's prediction and selecting the one with the smallest value. Intuitively, a prediction minimizes the objective by assigning low probability to classes similar to $i$ and high probability to those dissimilar to $i$.

**Extending LMA with HIPS.** Similarly, restricting the attack space of LMA to $\mathcal{A}$ prevents us from directly solving (2). However, we can show in Corollary 7, Appendix E.2 that (2) admits a solution at the extreme points of $\mathcal{A}$, which correspond to the honest predictions. Thus, we can compute the loss for each honest prediction and select the one that maximizes (2).

Table 2: FedDistill: 20 clients of which nine are byzantine ($\alpha = 0.45$). We report the final test accuracy averaged over multiple runs with standard deviation for different attacks (corresponding to rows) and defences (corresponding to columns). BA refers to the baseline accuracy, i.e., the final accuracy of FedDistill if all clients are honest. Recall that the goal of HIPS is to disrupt the learning process, hence lower accuracy is better.

| | CINIC-10 (ResNet-18), BA=80.2±0.1 | | | | CIFAR-100 (WideResNet-28), BA=66.8±0.5 | | | |
| --- | --- | --- | --- | --- | --- | --- | --- | --- |
| | Mean | GM | Cronus | EG+F | Mean | GM | Cronus | EG+F |
| LMA | 54.6±1.2 | 75.0±0.6 | 71.1±1.7 | 77.4±1.1 | 41.8±4.4 | 51.3±0.1 | 44.6±0.2 | 57.2±1.2 |
| CPA | 45.9±0.4 | 71.2±5.2 | 65.9±3.3 | 79.2±0.2 | 43.3±1.2 | 56.7±0.9 | 55.3±0.3 | 62.1±1.4 |
| HIPS+LMA | 75.3±0.1 | 68.7±0.1 | 67.7±1.0 | 73.3±0.9 | 50.3±3.3 | 34.3±0.4 | 34.4±2.8 | 49.3±0.5 |
| HIPS+CPA | 74.2±1.1 | 65.8±0.5 | 66.4±0.1 | 72.9±0.7 | 47.2±4.2 | 32.6±4.5 | 28.1±0.5 | 46.4±0.0 |

Table 2 shows that HIPS *improves* the effectiveness of CPA and LMA against EG+F, GM and Cronus but *reduces* its effectiveness against the mean. Recall that the objective of HIPS is to reduce the strength of attacks in order to make it harder to detect. Therefore, the fact that HIPS reduces the effectiveness of the attacks against the mean is to be expected as in the absence of a defence mechanism, making an attack less detectable does not have any advantage.

In consequence, the mean is slightly more resilient to HIPS attacks than EG+F, while EG+F is more resilient towards non-HIPS attacks by a large margin. Since we don't know in advance which attack strategy byzantine clients will use, the best defence is the one that experiences the smallest drop in accuracy against the most effective attack. This ensures that, regardless of which attack is employed, the defence performs reliably and minimizes the worst possible impact on model accuracy

## 6 DISCUSSION

We analyzed to what extent byzantine clients can perturb KD-based FL, exemplified by the prototypical algorithm FedDistill, allowing us to better understand and characterize its byzantine resilience.

By proposing two effective new attacks specifically for FedDistill, namely LMA and CPA, we highlighted vulnerabilities of FedDistill. We remark that while FedAVG attacks can not be directly applied to FedDistill, attacks for FedDistill can in turn always be applied to FedAVG by computing gradients based on the byzantine predictions. In fact, attacks based on computing gradients based on simple label-flipping attacks have been considered in the FedAVG literature (Allen-Zhu et al., 2021). Applying our attacks to FedAVG might be an interesting direction for further research.

To increase the reliability of KD-based FL, we introduced the novel defence mechanism ExpGuard, which we demonstrated to be more resilient against previously proposed attacks as well as our new attacks. Since ExpGuard does not have a significant impact on the accuracy of FedDistill in the absence of byzantine clients, does not require hyperparameter tuning and can be efficiently computed, it is a promising method to reduce the impact of arbitrary failures when using KD-based FL methods.

To obfuscate attacks by navigating the tradeoff between their strength and detectability, we proposed HIPS. Using this method, our attacks are able to reduce the accuracy of FedDistill even when using ExpGuard. We hope that our contributions lay the groundwork for future research in the domain of byzantine FL.

ACKNOWLEDGMENTS

This research was partially supported by the DFG Cluster of Excellence MATH+ (EXC-2046/1, project id 390685689) funded by the Deutsche Forschungsgemeinschaft (DFG) as well as by the German Federal Ministry of Education and Research (fund number 01IS23025B). Most of the notations in this work have a link to their definitions, using this code.

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

## A  BROADER IMPACTS

FL has the potential to harness previously inaccessible data, overcoming privacy concerns and which could have positive impact in domains like healthcare. This underscores the necessity of enhancing our comprehension of these methods. However, the adoption of distributed training setups introduces new challenges and risks.

Studying FL in the byzantine setting is often motivated as an avenue to study the resilience of these methods to network errors or bugs, the increasing deployment of FL in critical real-world contexts amplifies the urgency of addressing threats posed by adversarial actors. The notable case of the "Google Maps Hacks", an art performance by Simon Weckert, serves as a compelling illustration. Weckert simulated a virtual traffic jam in Google Maps by strolling around Berlin with 99 smartphones, manipulating the digital representation of streets. This demonstration had tangible consequences in the physical world, rerouting cars to avoid perceived traffic congestion. Extrapolating such exploits to critical infrastructure, like a smart grid reliant on usage information for energy production, underscores the potential ramifications of malicious interventions.

## B  EXPERIMENTAL DETAILS

**Public datasets**    For CIFAR-10, we use the unlabeled split of the STL-10 dataset (Coates et al., 2011), consisting of 100k samples. For CINIC-10 we use the validation split of CINIC-10 consisting of 90k samples. For Clothing1M, we use the noisily labeled split of Clothing1M, consisting of 1M samples.

For CIFAR-100, we use the train split both for the public dataset and as the private datasets. While this does not reflect a realistic use-case, we did not find an unlabeled dataset that was sufficiently in-domain and large enough to achieve good performance. We explored two alternative public datasets for CIFAR-100: Imagenet32 (Chrabaszcz et al., 2017), a downscaled version of the Imagenet dataset (Russakovsky et al., 2014) and an extended version of the CIFAR-100 dataset consisting of 30k images, found on Kaggle [1]. Using both of these datasets, we were not able to achieve good performance, which is probably due to domain drift. Since the goal of this paper is not to achieve high accuracy using KD-based FL methods but rather to evaluate byzantine attacks and defences we chose to use the same data for the local and public data. Alternatively, one could try to generate a public dataset using either a pretrained generative model or by training a generative model in via FL, see e.g. (Zhu et al., 2021).

**FedDistill training.**    To achieve good performance with FedDistill, we reinitialize the server model and train it to convergence in every communication round. Without re-initializing, we were not able to achieve high server test accuracy even with longer training. This aligns with the observation made by Achille et al. (2018) and Ash & Adams (2020) that using noisy data in the beginning of training permanently damages the model, even if the data quality improves later on. In the early communication rounds, the clients predictions on the public dataset are inaccurate. Hence, even if the clients models were to improve over time, the early training on inaccurate labels prevents the server from learning an accurate classifier. By re-initializing the server, this problem can be circumvented and the server can learn from the progressively improving clients predictions.

**Hyperparameters**    We run the experiments with two different random seeds and provide the standard deviation. In each communication round, the server collects the soft predictions of the clients, i.e., the probability distribution over the classes, and aggregates them using either the mean or other aggregation methods. The server then re-initializes the model and trains it on the public dataset using the aggregated predictions as a target. In each round, it uses SGD+momentum with a linearly decaying learning rate. Then, the clients train the model on their private data. The clients also use SGD with momentum and a linearly decaying learning rate. However, their linear decay stretches across the total epochs, meaning that they do not restart the schedule in each communication round as the server. For both the server and clients, we calculate the accuracy on the validation dataset after each epoch and implement early stopping to prevent overfitting.

---

[1] https://www.kaggle.com/datasets/dunky11/cifar100-100x100-images-extension/data

|                         | CIFAR-10  | CIFAR-100     | CINIC-10  | Clothing1M |
|-------------------------|-----------|---------------|-----------|------------|
| Architecture            | ResNet-18 | WideResNet-28 | ResNet-18 | ResNet-50  |
| Batch size              | 128       | 256           | 128       | 256        |
| Server epochs per round | 80        | 100           | 80        | 10         |
| Total local epochs      | 400       | 250           | 200       | 300        |

## C  FURTHER RESULTS.

Table 3 shows the effect of different attacks and defences in the same setting as in Table 1 and Table 2 but for different datasets.

Table 3: FedDistill: 20 clients of which nine are byzantine ($\alpha = 0.45$). Final test accuracy averaged over multiple runs with standard deviation for different attacks and defences. BA refers to the baseline accuracy, i.e., the final accuracy of FedDistill if all clients are honest.

|          | CIFAR-10 (ResNet-18), BA=87.7±1.2 | | | | Clothing1M (ResNet-50), BA=69.0±0.3 | | | |
|----------|----------|----------|-----------|----------|----------|----------|----------|----------|
|          | Mean     | GM       | Cronus    | EG+F     | Mean     | GM       | Cronus   | EG+F     |
| RLF      | 69.4±1.2 | 68.7±0.8 | 68.6±0.4  | 68.7±1.1 | 84.6±0.1 | 85.4±0.6 | 84.7±0.3 | 85.4±0.0 |
| LMA      | 40.3±3.3 | 58.3±0.7 | 61.4±0.5  | 68.3±0.7 | 73.4±8.6 | 83.3±0.2 | 80.6±2.3 | 85.4±0.1 |
| CPA      | 33.7±2.7 | 58.4±0.3 | 43.9±12.9 | 68.5±1.0 | 68.4±0.8 | 78.4±0.9 | 74.5±0.6 | 85.5±0.8 |
| HIPS+LMA | 33.7±2.7 | 58.4±0.3 | 43.9±12.9 | 68.5±1.0 | 84.8±0.1 | 78.0±1.6 | 78.5±1.1 | 83.8±0.2 |
| HIPS+CPA | 63.4±    | 55.2±1.1 | 54.8±2.1  | 57.7±0.5 | 85.0±0.1 | 79.4±0.8 | 77.3±0.1 | 83.2±0.9 |

**Performance of FedDistill and FedAVG in the benign setting.**   We compare the performance of FedDistill and FedAVG in the benign setting. The experimental setup for FedDistill follows the same configuration as in Table 1, utilizing 20 clients, 10 communication rounds, but without byzantine clients. For FedAVG, we follow this setup as closely as possible, however for CIFAR-100 and Clothing1M, we had to use 100 communication rounds to achieve competitive performance.

The reason for the discrepancy between FedDistill and FedAVG on Clothing1M is that FedAVG is does not utilize the unlabeled public dataset. This highlights the fact that FedAVG and FedDistill make different assumptions about the data.

Table 4: Comparison of final test accuracy between FedAVG and FedDistill across different datasets. Results are averaged over multiple runs with standard deviation included.

|            | CIFAR-10 | CINIC-10 | CIFAR-100 | Clothing1M |
|------------|----------|----------|-----------|------------|
| FedAVG     | 88.6±0.1 | 75.1±0.2 | 67.2±0.3  | 61.2±1.1   |
| FedDistill | 86.8±1.0 | 80.2±0.1 | 66.8±0.5  | 69.0±0.3   |

**Performance of defenses in benign settings.**   We evaluate the performance of different defense mechanisms in the benign setting. The results, as detailed in Table 5, demonstrate that the accuracy of the defenses remains largely consistent with the mean baseline, with only minor reductions in accuracy. These experiments were conducted without altering any hyperparameters, indicating that the slight accuracy variations observed could potentially be minimized through hyperparameter tuning.

**Comparison to state-of-the-art FedAVG defence mechanisms.**   We implemented the state-of-the-art approach of using D-SGD with momentum and nearest neighbour mixing (NNM) with coordinate-wise median (CWMED) aggregation for CINIC-10 against the ALIE attack, as outlined in Allouah et al. (2023). In the absence of attacks, D-SGD achieves an accuracy of 80.8 with mean

Table 5: Performance of FedDistill with different defense strategies in the benign setting without byzantine clients. Results are averaged over multiple runs with standard deviation included.

|                    | CIFAR-10 | CINIC-10 | CIFAR-100 | Clothing1M |
|--------------------|----------|----------|-----------|------------|
| FedDistill (E+F)   | 86.2±1.0 | 80.5±0.2 | 64.9±1.9  | 68.0±0.2   |
| FedDistill (GM)    | 87.0±0.1 | 81.1±0.5 | 66.4±0.4  | 69.4±0.0   |
| FedDistill (Cronus)| 86.4±0.3 | 80.4±0.2 | 62.6±0.8  | 64.9±0.2   |
| FedDistill (mean)  | 87.7±1.2 | 80.2±0.1 | 66.8±0.5  | 69.0±0.3   |

aggregation and 77.7 with NNM+CWMED. Table 6 presents the accuracy for varying the number of byzantine clients and attack magnitudes $\eta$. Note that a direct byzantine comparison between FedDistill and FedAVG is hard to interpret, as both methods make different assumptions about the data and have different advantages and disadvantages.

Byzantine clients can still significantly disrupt the learning process, reducing accuracy to below 20% with 9 byzantine clients. This represents a more than fourfold decrease in accuracy compared to the scenario without byzantine clients, whereas FedDistill only experiences a 1.1-fold reduction. We have not optimized the attack amplitude for ALIE, as suggested by Allouah et al. (2023), which means that these results are conservative and optimal attack amplitudes might lead to even lower accuracies.

Table 6: Accuracy of D-SGD under ALIE attack with different amounts of byzantine clients and attack magnitudes $\eta$.

| # byz. clients \| $\eta$ | 0.05      | 0.1      | 0.5      | 1        | 2         |
|--------------------------|-----------|----------|----------|----------|-----------|
| 1                        | 78.7±0.4  | 78.3±0.7 | 69.0±2.4 | 75.8±0.0 | 79.9±0.0  |
| 5                        | 77.2±0.8  | 70.0±4.0 | 27.6±1.5 | 24.1±0.2 | 71.5±0.0  |
| 9                        | 55.8±26.0 | 55.3±5.8 | 25.8±2.9 | 19.8±0.2 | 62.2±13.7 |

# D  ABLATIONS

**Computational cost of aggregation methods**    In Table 7, we show the runtime of GM and EG+F for multiple datasets. Even for large datasets with many classes, the computational overhead is negligible compared to the cost of training the models. Note that the runtime is for the whole dataset. This means that for the experimental setting we consider throughout the paper with 10 communication rounds, the total time spent on aggregating the predictions is about 32 seconds, even for CIFAR-100.

Table 7: Runtime per communication round of different defence methods for datasets of different size and with different number of classes.

|                | STL-10 | CIFAR-100 | Clothing1M |
|----------------|--------|-----------|------------|
| Samples        | 100k   | 60k       | 1M         |
| Classes        | 10     | 100       | 14         |
| Runtime: EG+F  | 0.08s  | 3.2s      | 0.8s       |
| Runtime: GM    | 0.15s  | 0.62s     | 0.77s      |

**ExpGuard with other defences.**    Table 8 compares the performance of different robust aggregation methods combined with ExpGuard. The resilience of all aggregation methods is significantly improved by ExpGuard, cf. Tables 1 and 2. Further, EG+F clearly outperforms ExpGuard combined with the other aggregation methods while ExpGuard+Cronus is the *least* robust among all methods, except for ExpGuard+Mean. This shows that the modifications of EG+F are crucial to achieve good performance.

Table 8: FedDistill: 20 clients of which nine are byzantine ($\alpha = 0.45$). Final test accuracy averaged over multiple runs with standard deviation for different attacks and defences used **in combination with ExpGuard**. BA refers to the baseline accuracy, i.e., the final accuracy of FedDistill if all clients are honest.

| | CINIC-10 (ResNet-18), BA=80.2±0.1 | | | | CIFAR-10 (ResNet-18), BA=87.7±1.2 | | | |
|---|---|---|---|---|---|---|---|---|
| | Mean | GM | Filter | Cronus | Mean | GM | Filter | Cronus |
| RLF | 77.1 ±0.4 | 79.3 ±0.9 | 78.8±0.1 | 76.3 ±0.2 | 77.1 ±0.4 | 86.0 ±0.3 | 85.4±0.0 | 85.9 ±0.8 |
| LMA | 53.0 ±4.8 | 73.8 ±2.0 | 77.4±1.1 | 71.2 ±1.3 | 53.0 ±4.8 | 84.1 ±0.9 | 85.4±0.1 | 51.5 ±14.9 |
| CPA | 45.8 ±0.7 | 71.9 ±0.6 | 79.2±0.2 | 66.1 ±0.5 | 45.8 ±0.7 | 10.7 ±18.2 | 85.5±0.8 | 2.3 ±0.4 |
| HIPS+LMA | 75.1 ±0.8 | 67.9 ±0.7 | 73.3±0.9 | 67.5 ±0.9 | 75.1 ±0.8 | 76.5 ±0.9 | 83.8±0.2 | 76.5 ±0.8 |
| HIPS+CPA | 72.7 ±0.4 | 66.2 ±0.2 | 72.9±0.7 | 65.7 ±0.0 | 72.7 ±0.4 | 77.5 ±0.8 | 83.2±0.9 | 76.9 ±0.4 |

**Number of clients and communication rounds**    In Figure 5, we ablate the impact of the number of clients and communication rounds on FedDistill in the honest setting, while keeping the other parameters such as the total number of local epochs and server training epochs per round fixed.

On the left side of Figure 5, we see that increasing the number of clients leads to decreasing accuracy. This is due to the fact that the datasets we use are of fixed size and so the amount of private data per client decreases as the number of clients increases. This is especially problematic before the first communication round, as the models have to train a model from scratch that encodes some useful information only based on their private dataset.

On the right side of Figure 5, we see that at first, increasing the number of communication rounds leads to an improvement, but as the number of communication rounds decreases, we first experience diminishing returns at some point even a decrease in accuracy. This in part due to the fact that we kept the total number of local epochs fixed, but we observed similar result when increasing the total number of local epochs.

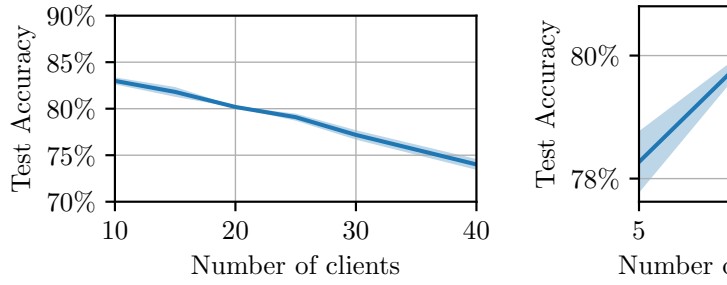 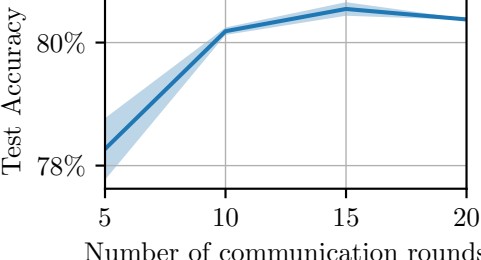

Figure 5: (Left) ResNet-18 on CINIC-10 with 10 communication rounds: Final test accuracy of FedDistill with different number of clients. (Right) ResNet-18 on CINIC-10 with 20 clients: Final test accuracy of FedDistill with different number of communication rounds.

**Measuring the impact of byzantine clients on EG+F performance.**    We conducted an ablation study to assess the impact of varying the number of byzantine clients on the performance of EG+F on the CINIC-10 dataset, see Table 9. We selected LMA with and without HIPS as the attack strategy, as it was the most effective attack on CINIC-10 with 9 byzantine clients. For reference, the baseline performance of FedDistill using mean aggregation without byzantine clients is 80.2.

**Evaluating byzantine robustness on non-i.i.d. data.**    We evaluate the performance of FedDistill in the non-i.i.d. setting. To that end, we generate non-i.i.d. data using the Dirichlet distribution with parameter $\beta$, which is a common approach in the FL literature (Lin et al., 2020). We follow the same setting as in Table 1, except that the client datasets do not follow a uniform class distribution, but rather a skewed one depending on the parameter $\beta$, where smaller $\beta$ means more heterogeneity and vice-versa. In Table 10 we provide experimental results for the CINIC-10 and CIFAR-10 datasets.

Table 9: Test accuracy of EG+F on CINIC-10 with varying numbers of byzantine clients. Results are averaged over multiple runs with standard deviation included.

| # Byzantine Clients | 0 | 3 | 5 | 7 | 9 |
|---|---|---|---|---|---|
| HIPS+LMA | 80.5±1.2 | 79.7±1.0 | 79.8±0.4 | 78.9±0.1 | 72.9±0.7 |
| LMA | 80.5±1.2 | 79.7±0.1 | 79.9±0.0 | 79.9±0.2 | 77.4±1.1 |

We conducted a comprehensive evaluation of all attack and defense strategies, varying the $\beta$ in $\{10, 5, 1\}$. The table differentiates the defense mechanisms, presenting the final test accuracies for various attacks (listed as rows for each defense) in both the i.i.d. and non-i.i.d. scenarios with different $\beta$ values. Notably, for attacks without HIPS, EG+F demonstrates superior resilience compared to all other defenses, with the disparity being even more pronounced in the non-i.i.d. setting, particularly at lower $\beta$ values. Specifically, for $\beta = 1$, both GM and Cronus underperform relative to the mean, whereas EG+F achieves more than double the accuracy of the mean. Unlike other defense methods, EG+F retains its robustness amidst higher heterogeneity, making it an ideal choice for non-i.i.d. byzantine-resilient aggregation.

As in the i.i.d. scenario, the mean shows slightly better robustness against HIPS attacks than EG+F. However, the mean lacks resilience against non-HIPS attacks. Consequently, if the server is uncertain about the type of attacks it might face, EG+F remains the preferable choice, offering solid worst-case performance with only a minor reduction in performance against HIPS attacks. Generally, HIPS proves to be significantly more potent in the non-i.i.d. setting than in the i.i.d. one. This is expected, as increased heterogeneity in client data results in more heterogeneous predictions increasing the size of the attack space, i.e., the convex hull of honest client predictions. This allows the byzantine clients to more easily evade detection while disrupting the server.

Table 10: Test accuracy of different defense mechanisms on CINIC-10 and CIFAR-10 datasets with varying non-i.i.d. levels ($\beta$). Results are averaged over multiple seeds, but we omitted standard deviations for the sake of legibility. The table is divided into sections for each defense mechanism.

| | CINIC-10 | | | | CIFAR-10 | | | |
|---|---|---|---|---|---|---|---|---|
| | i.i.d. | $\beta = 10$ | $\beta = 5$ | $\beta = 1$ | i.i.d. | $\beta = 10$ | $\beta = 5$ | $\beta = 1$ |
| **Mean** | | | | | | | | |
| RLF | 76.9 | 75.5 | 73.9 | 62.0 | 84.6 | 83.9 | 80.6 | 71.0 |
| CPA | 45.9 | 37.2 | 28.1 | 18.5 | 68.4 | 57.9 | 38.2 | 15.4 |
| LMA | 54.6 | 52.0 | 47.4 | 27.2 | 73.4 | 69.5 | 65.6 | 35.5 |
| HIPS+CPA | 74.2 | 60.7 | 53.4 | 32.6 | 85.0 | 76.2 | 59.5 | 32.3 |
| HIPS+LMA | 75.3 | 61.2 | 52.8 | 28.9 | 84.9 | 75.6 | 54.0 | 32.0 |
| **EG+F** | | | | | | | | |
| RLF | 78.8 | 75.8 | 73.0 | 63.8 | 85.4 | 83.8 | 80.3 | 69.1 |
| CPA | 73.3 | 77.9 | 73.6 | 58.4 | 85.5 | 85.6 | 83.5 | 52.9 |
| LMA | 77.4 | 75.8 | 73.8 | 62.0 | 85.4 | 85.0 | 83.4 | 74.0 |
| HIPS+CPA | 72.9 | 57.4 | 41.4 | 24.3 | 57.7 | 68.3 | 42.8 | 25.3 |
| HIPS+LMA | 73.4 | 59.1 | 44.0 | 24.4 | 68.5 | 68.2 | 44.6 | 23.0 |
| **GM** | | | | | | | | |
| RLF | 79.3 | 76.3 | 75.2 | 62.5 | 85.4 | 73.1 | 82.4 | 85.2 |
| CPA | 71.2 | 44.5 | 37.6 | 9.5 | 78.4 | 67.4 | 43.8 | 10.4 |
| LMA | 75.0 | 55.3 | 47.2 | 10.4 | 83.3 | 75.0 | 58.7 | 9.6 |
| HIPS+CPA | 65.8 | 38.6 | 31.4 | 18.4 | 55.2 | 59.7 | 30.2 | 13.5 |
| HIPS+LMA | 68.7 | 36.9 | 30.1 | 15.1 | 58.4 | 45.6 | 28.2 | 10.0 |
| **Cronus** | | | | | | | | |
| RLF | 76.6 | 70.8 | 62.6 | 39.1 | 84.7 | 79.4 | 75.3 | 42.7 |
| CPA | 65.9 | 40.1 | 25.7 | 9.1 | 74.5 | 60.0 | 34.9 | 10.1 |
| LMA | 71.1 | 51.4 | 35.1 | 10.0 | 80.6 | 71.8 | 48.0 | 7.8 |
| HIPS+CPA | 66.4 | 38.3 | 29.5 | 17.4 | 54.8 | 48.1 | 28.9 | 16.8 |
| HIPS+LMA | 67.7 | 37.3 | 28.6 | 16.1 | 43.9 | 42.7 | 21.0 | 10.4 |

# E    THEORETICAL RESULTS

In this section, we present the formal statements and proof discussed in the main part of the paper.

## E.1    BOUNDING THE BYZANTINE ERROR

First we show in Lemma 2 that for the two most common loss functions, namely the MSE and CEL, the gradient of the Loss with respect to the parameters of the classifier $w$ are linear with respect to the difference between the models prediction and the target label and it's slope is defined only by the model parameters $w$ and the data $x$. In Theorem 3, we show based on Lemma 2 that if a parameter $\tilde{w}$ is a stationary point of ($\mathcal{P}_{\text{distill}}$), then it is also in the neighbourhood of an approximate stationary point of ($\mathcal{P}_{\text{honest}}$). The size of this neighbourhood is determined by the fraction of byzantines $\alpha$ and a constant $C$ depending on the data $x \in \mathcal{D}_{\text{pub}}$ and $\tilde{w}$. In particular as $\alpha$ goes to zero the size of the neighbourhood goes to zero as well.

**Lemma 2.** *Consider a function $\mathcal{L}(h(x, w), y)$ where $h : \mathcal{X} \times \mathcal{W} \to \Delta_c$ such that $w \mapsto h(x, w)$ is differentiable and $y \in \Delta_c$. Assume that one of the two statements hold*

   *1. $\mathcal{L}(a, b) = \text{MSE}(a, b)$*

   *2. $\mathcal{L}(a, b) = \text{CEL}(b, a)$ and $h(x, w) = \sigma \circ \Phi$ where $\Phi : \mathcal{X} \times \mathcal{W} \to \mathbb{R}^c$ and $\sigma$ is the softmax function.*

*Then there is a linear function $g : \Delta_c \to \mathcal{W}$ such that $g(h - y) = \nabla_w \mathcal{L}(h(x, w), y)$ which is $L_p$-Lipschitz, where $L_p$ depends on $x$ and $w$.*

*Proof of Lemma 2.*

**Case 1.** We have $\text{MSE}(p, q) = \frac{1}{2} \|p - q\|^2$. Then using the shorthand $h_i$ for the $i$-th element of $h(x, w)$ and by applying the chain rule, we have

$$\frac{\partial \text{MSE}(h(x, w), y)}{\partial w_j} = \sum_{i=1}^{c} \frac{\partial \text{MSE}(h(x, w), y)}{\partial h_i} \frac{\partial h_i}{\partial w_j} = \sum_{i=1}^{c} (y_i - h_i) \frac{\partial h_i}{\partial w_j}.$$

Hence we can write the gradient as follows

$$\nabla_w \text{MSE}(h(x, w), y) = J_h(w)(y - h(x, w)),$$

where $J_h$ is the Jacobian of $h$. Let $g(h(x, w) - y) \stackrel{\text{def}}{=} J_h(w)^T (y - h(x, w))$. Then $g$ is linear and $L_p$-Lipschitz, where $L_p \leq \|J_h(w)\|$. Note that $J_h(w)$ depends only on $x \in \mathcal{D}_{\text{pub}}$ and $w$.

**Case 2.** We have that $\text{CEL}(p, q) \stackrel{\text{def}}{=} -\sum_{i=1}^{n} p_i \log(q_i)$ and $\nabla_q \text{CEL}(p, q) = -(p_1/q_1, \ldots, p_N/q_N)^T$. Furthermore, $\sigma_i(p) \stackrel{\text{def}}{=} \frac{\exp(p_i)}{\sum_{i=1}^{n} \exp(p_i)}$ and $\frac{\partial \sigma_i(p)}{\partial p_j} = \sigma_i(p)(\delta_{ij} - \sigma_j(p))$ where $\delta_{ij} = 1$ if $i = j$ and $\delta_{ij} = 0$ otherwise.

$$\frac{\partial \sigma_i(p)}{\partial p_j} = \sigma_i(p)(\delta_{ij} - \sigma_j(p)), \quad \text{with } \delta_{ij} = \begin{cases} 1, & i = j \\ 0, & \text{otherwise.} \end{cases}$$

Hence, using the chain rule,

$$\frac{\partial \text{CEL}(y, \sigma(\Phi))}{\partial \Phi_i} = \sum_{j=1}^{c} \frac{\partial \text{CEL}(y, \sigma(\Phi))}{\partial \sigma_j(\Phi)} \cdot \frac{\partial \sigma_j(\Phi)}{\partial \Phi_i} = \sum_{j=1}^{c} (y_j/\sigma_j(\Phi)) \sigma_j(\Phi)(\delta_{ij} - \sigma_i(\Phi))$$

$$= \sum_{j=1}^{c} y_j (\delta_{ij} - \sigma_i(\Phi)) = -\sigma_i(\Phi) \left( \sum_{j=1}^{c} y_j \right) + y_i = y_i - \sigma_i(\Phi).$$

Then,

$$\frac{\partial \text{CEL}(y, \sigma(\Phi))}{\partial w_j} = \sum_{i=1}^{c} \frac{\partial \text{CEL}(y, \sigma(\Phi))}{\partial \Phi_i} \frac{\partial \Phi_i}{\partial w_j} = \sum_{i=1}^{c} (y_i - \sigma_i(\Phi)) \frac{\partial \Phi_i}{\partial w_j}.$$

It follows that

$$\nabla_w \text{CEL}(y, \sigma(\Phi)) = J_\Phi(w)^T (y - h(x, w)),$$

where $J_h$ is the Jacobian of $h$. Now let $g(h(x, w) - y) \stackrel{\text{def}}{=} J_\Phi(w)^T (y - h(x, w))$. Then $g$ is linear and $L_p$-Lipschitz, where $L_p \leq \|J_h(w)\|$. Note that $J_h(w)$ depends only on $x \in \mathcal{D}_{\text{pub}}$ and $w$.

□

**Theorem 3.** *Consider the optimization problem defined in ($\mathcal{P}_{\text{honest}}$), i.e.,*

$$\min_{w \in \mathcal{W}} \left\{ F(w) \stackrel{\text{def}}{=} \frac{1}{|\mathcal{D}_{pub}|} \sum_{x \in \mathcal{D}_{pub}} \mathcal{L}(h(x, w), \bar{Y}_{\mathcal{H}}(x)) \right\}.$$

*and the optimization problem defined in ($\mathcal{P}_{\text{distill}}$), i.e.,*

$$\min_{w \in \mathcal{W}} \left\{ \tilde{F}(w) \stackrel{\text{def}}{=} \frac{1}{|\mathcal{D}_{pub}|} \sum_{x \in \mathcal{D}_{pub}} \mathcal{L}(h(x, w), \bar{Y}(x)) \right\}.$$

*Assume that one of the following holds:*

1. $\mathcal{L}(p, q) = \text{CEL}(q, p)$ and $h(x, w) = \sigma \circ \Phi$ with function $\Phi : \mathcal{X} \times \mathcal{W} \to \mathbb{R}^c$ and $\sigma$ is the softmax function.

2. $\mathcal{L}(p, q) = \text{MSE}(p, q)$.

Then, if $\tilde{w}$ is a stationary point of $\tilde{F}$, i.e., $\left\|\nabla \tilde{F}(\tilde{w})\right\| = 0$, it also holds that

$$\|\nabla F(\tilde{w})\| \leq \frac{1}{|\mathcal{D}_{pub}|} \sum_{x \in \mathcal{D}_{pub}} C\alpha \left\|\bar{Y}_{\mathcal{H}}(x) - \bar{Y}_{\mathcal{B}}(x)\right\| \leq \sqrt{2}\alpha C,$$

where $C > 0$ is a constant which depends only on $\mathcal{D}_{pub}$ and $w$, and is independent of the client predictions. If further, $\mathcal{L}(h(x, \cdot), w)$ is L-smooth, then running SGD on $\tilde{F}$ initialized at $w_0$ for $T = \mathcal{O}\left(\frac{L^2 F(w_0)^2 + \sigma^4}{\varepsilon^2}\right)$ iterations, where $\sigma$ is the gradient variance, then it holds that $\|\nabla F(\bar{w}_T)\|^2 = \mathcal{O}(\varepsilon + C^2\alpha^2)$.

*Proof of Theorem 3.* We prove the theorem in two parts. First we bound the norm of the gradient error and then we show approximate convergence of SGD given the gradient error bound. Note that $\nabla F(w)$ and $\nabla \tilde{F}(w)$ only differ in the second argument of $\mathcal{L}$. By Lemma 2, we have that

$$\nabla F(w) = \frac{1}{|\mathcal{D}_{\text{pub}}|} \sum_{x \in \mathcal{D}_{\text{pub}}} \nabla \mathcal{L}(h(x, w), \bar{Y}_{\mathcal{H}}(x)), \quad \nabla \tilde{F}(w) = \frac{1}{|\mathcal{D}_{\text{pub}}|} \sum_{x \in \mathcal{D}_{\text{pub}}} \nabla \mathcal{L}(h(x, w), \bar{Y}(x)),$$

are $C$-Lipschitz function with respect to the second argument, where $C$ depends on $w$ and $x \in \mathcal{D}_{\text{pub}}$ only. Denote by $\|B(w)\| \overset{\text{def}}{=} \left\|\nabla F(w) - \nabla \tilde{F}(w)\right\|$ the gradient error, then

$$\|B(w)\| \leq \sum_{x \in \mathcal{D}_{\text{pub}}} \frac{1}{|\mathcal{D}_{\text{pub}}|} \left\|\bar{Y}_{\mathcal{H}}(x) - \bar{Y}(x)\right\| = \sum_{x \in \mathcal{D}_{\text{pub}}} \frac{\alpha C}{|\mathcal{D}_{\text{pub}}|} \left\|\bar{Y}_{\mathcal{H}}(x) - \bar{Y}_{\mathcal{B}}(x)\right\| \leq \sqrt{2}\alpha C, \quad (3)$$

where the first inequality holds by the Lipschitzness, the second inequality holds since $\bar{Y} = (1 - \alpha)\bar{Y}_{\mathcal{H}} + \alpha\bar{Y}_{\mathcal{B}}$ and the last inequality holds since the maximum $\ell_2$ distance between two points in $\Delta_c$ is $\sqrt{2}$. Now let $\tilde{w}$ be a stationary point of $\tilde{F}$, then we have

$$\|\nabla F(\tilde{w})\| = \left\|\nabla F(\tilde{w}) - \nabla \tilde{F}(\tilde{w})\right\| = \|B(\tilde{w})\|,$$

which together with Equation (3) shows the first part of the theorem.

For the second part of the proof, we define

$$f_i(w) = \mathcal{L}(h(x_i, w), \bar{Y}_{\mathcal{H}}(x_i)), \quad \tilde{f}_i(w) = \mathcal{L}(h(x_i, w), \bar{Y}(x_i))$$

and it follows that

$$F(w) = \mathbb{E}_i[f_i(w)] = \frac{1}{|\mathcal{D}_{\text{pub}}|} \sum_{i=1}^{|\mathcal{D}_{\text{pub}}|} f_i(w) \text{ and } \tilde{F}(w) = \mathbb{E}_i[\tilde{f}_i(w)] = \frac{1}{|\mathcal{D}_{\text{pub}}|} \sum_{i=1}^{|\mathcal{D}_{\text{pub}}|} \tilde{f}_i(w).$$

The update rule of SGD is defined as $w_{t+1} \leftarrow w_t - \gamma \nabla \tilde{f}_i(w_t)$, where $i$ is sampled uniformly at random. We have that

$$\nabla \tilde{f}_i(w_t) = \nabla F(w_t) + \nabla \tilde{f}_i(w_t) - \nabla f_i(w_t) + \nabla f_i(w_t) - \nabla F(w_t)$$

By the $L$ smoothness of $\mathcal{L}(h(x, \cdot), y)$ and the SGD update rule, we have

$$f_i(w_{t+1}) - f_i(w_t) \leq \langle \nabla f_i(w_t), w_{t+1} - w_t \rangle + \frac{L}{2} \|w_{t+1} - w_t\|^2$$

$$= -\gamma \langle \nabla f_i(w_t), \nabla \tilde{f}_i(w_t) \rangle + \frac{L\gamma^2}{2} \left\|\nabla \tilde{f}_i(w_t)\right\|^2.$$

Now, taking the expectation with respect to $i$ on both sides, we obtain

$$F(w_{t+1}) - F(w_t) \leq -\gamma\langle\nabla F(w_t), \nabla\tilde{F}(w_t)\rangle + \frac{L\gamma^2}{2}\mathbb{E}\left[\left\|\nabla\tilde{f}_i(w_t)\right\|^2\right]$$

$$= -\gamma\langle\nabla F(w_t), \nabla\tilde{F}(w_t)\rangle + \frac{L\gamma^2}{2}\left(\mathbb{E}\left[\left\|\nabla\tilde{f}_i(w_t) - \nabla\tilde{F}(w_t)\right\|^2\right] + \left\|\nabla\tilde{F}(w_t)\right\|^2\right),$$

where the second inequality holds by noting that $\mathbb{E}[X^2] = \text{Var}(X) + \mathbb{E}[X]^2$. We bound,

$$\mathbb{E}\left[\left\|\nabla\tilde{f}_i(w_t) - \nabla\tilde{F}(w_t)\right\|^2\right] = \mathbb{E}\left[\left\|\nabla\tilde{f}_i(w_t) - \nabla f_i(w_t) + \nabla f_i(w_t) - \nabla F(w_t) - \nabla F(w_t) + \nabla\tilde{F}(w_t)\right\|^2\right]$$

$$\leq 3\mathbb{E}\left[\left\|\nabla\tilde{f}_i(w_t) - \nabla f_i(w_t)\right\|^2\right] + 3\mathbb{E}\left[\|\nabla f_i(w_t) - \nabla F(w_t)\|\right] + 3\|B(w_t)\|^2$$

$$\leq 6C^2\alpha^2 + 3\sigma^2 + 3\|B(w_t)\|^2$$

where the last line follows by noting $\sigma^2 \stackrel{\text{def}}{=} \mathbb{E}\left[\|\nabla f_i(w_t) - \nabla F(w_t)\|^2\right]$ and since

$$\mathbb{E}\left[\left\|\nabla\tilde{f}_i(w_t) - \nabla f_i(w_t)\right\|^2\right] = \frac{1}{|\mathcal{D}_{\text{pub}}|}\sum_{i=1}^{|\mathcal{D}_{\text{pub}}|}\left\|\tilde{f}_i(w_t) - \nabla f_i(w_t)\right\|^2 \leq 2C^2\alpha^2.$$

Using these terms, we further bound

$$F(w_{t+1}) - F(w_t) \leq -\gamma\langle\nabla F(w_t), \nabla\tilde{F}(w_t)\rangle + \frac{L\gamma^2}{2}\left(6C^2\alpha^2 + 3\sigma^2 + 3\|B(w_t)\|^2 + \left\|\nabla\tilde{F}(w_t)\right\|^2\right)$$

$$= -\gamma\mathbb{E}\left[\|\nabla f_i(w_t)\|^2\right] + 6\gamma C^2\alpha^2 + \frac{3L\gamma^2\sigma^2}{2},$$

where the last inequality follows by choosing $\gamma \leq \frac{2}{3L}$ and since by Equation (3), we have $\|B(w_t)\|^2 \leq 2\alpha^2 C^2$. Rearranging, summing from $t = 0, \ldots, T-1$ and dividing by $T$, we obtain

$$\frac{1}{T}\sum_{t=0}^{T-1}\mathbb{E}\left[\|\nabla f_i(w_t)\|^2\right] \leq \frac{F(w_0) - F(w_T)}{\gamma T} + 6\alpha^2 C^2 + \frac{3L\gamma\sigma^2}{2}$$

Then, choosing $\gamma = \frac{2}{3L\sqrt{T}}$, we obtain

$$\frac{1}{T}\sum_{t=0}^{T-1}\mathbb{E}\left[\|\nabla f_i(w_t)\|^2\right] \leq \frac{3LF(w_0)}{2\sqrt{T}} + 6\alpha^2 C^2 + \frac{\sigma^2}{\sqrt{T}}$$

Hence, choosing $T = \mathcal{O}(\frac{L^2 F(w_0)^2 + \sigma^4}{\varepsilon^2})$ ensures that

$$\frac{1}{T}\sum_{t=0}^{T-1}\mathbb{E}\left[\|\nabla f_i(w_t)\|^2\right] = \mathcal{O}\left(\varepsilon + \alpha^2 C_y^2\right).$$

Note that $\frac{1}{T}\sum_{t=0}^{T-1}\mathbb{E}\left[\|\nabla f_i(w_t)\|^2\right]$ is equal to $\mathbb{E}\left[\|\nabla f_i(\bar{w}_T)\|^2\right]$, where $\bar{w}_T$ is drawn uniformly at random from $\{w_0, \ldots, w_{T+1}\}$. This concludes the proof. $\qquad\square$

## E.2 ATTACK OPTIMALITY

In this subsection, we discuss the results concerning the attacks discussed in the paper. In Lemma 4, we prove that optimization problem satisfying certain conditions, of which the optimization problems corresponding to our attacks are instances, always admit an optimizer on the extreme point of the constraint set $\mathcal{X}$. Based on this result, we solve the optimization problem associated to (2) and its HIPS variant in closed form in Lemma 5 and Corollary 7.

**Lemma 4.** *Let $\mathcal{X} \subset \mathbb{R}^c$ be a convex polytope. Let $f : \mathbb{R}^c \to \mathbb{R}$ be continuous and convex in $\mathcal{X}$. Then the optimization problem*

$$\max_{x\in\mathcal{X}} f(x),$$

*admits an optimizer on an extreme point of $\mathcal{X}$.*

*Proof of Lemma 4.* By the extreme value theorem, $f$ attains a maximum $x^*$ in $\mathcal{X}$. Since $\mathcal{X}$ is a convex polytope, each $x \in \mathcal{X}$ can be represented as a convex combination of the extreme points of $\mathcal{X}$ denoted by $\text{Ext}(\mathcal{X})$. Denote by $\{x_i\}_{i \in [n]}$ the extreme points of $\mathcal{X}$. In particular we can represent the maximum as $x^* = \sum_{i=1}^{n} p_i x_i$ where $p \in \Delta_n$. By Jensen's inequality, we have

$$f(x^*) = f\left(\sum_{i=1}^{n} p_i x_i\right) \leq \sum_{i=1}^{n} p_i f(x_i)$$

Since all $p_i$ are non-negative, there exists at least one $x_i$ such that $f(x_i) \geq f(x^*)$, by the optimality of $x^*$, we have that $f(x^*) \geq f(x)$ for all $x \in \mathcal{X}$, hence $f(x_i) = f(x^*)$, which concludes the proof. $\qquad\square$

**Lemma 5.** *The prediction $\bar{Y}_{\mathcal{B}}^{\,*} = \mathbb{1}_i$ with $i \in \arg\min \bar{Y}_{\mathcal{H}}$ is an optimizer of* (2) *if one of the two statements hold*

1. $\mathcal{L}(a, b) = \text{CEL}(b, a)$ *and* $\bar{Y}_{\mathcal{H}} \in \text{Int}(\Delta_c)$

2. $\mathcal{L}(a, b) = \text{MSE}(a, b)$ .

**Remark 6.** *The image of the softmax function is* $\text{Int}(\Delta_c)$*. This means that if the honest clients use the softmax activation function in order to obtain a probability distribution over the classes, which is standard practice in training neural networks, the assumption* $\bar{Y}_{\mathcal{H}} \in \text{Int}(\Delta_c)$ *in Lemma 5 is always satisfied.*

*Proof of Lemma 5.* For the sake of simplicity, we write $y \overset{\text{def}}{=} \bar{Y}_{\mathcal{H}}(x)$ and $x \overset{\text{def}}{=} \bar{Y}_{\mathcal{B}}(x)$ throughout this proof.

**Case 1.** The MSE is convex and differentiable, hence by Lemma 4, there is an optimizer of Equation (2) on an extreme point of $\Delta_c$, i.e., the one-hot vector $\mathbb{1}_i$ for all $i \in [c]$. Using $\mathcal{L} = \text{MSE}$ and the characterization of the optimizer, we rewrite Equation (2) as follows

$$\max_{x \in \Delta_c} \frac{1}{2} \|y - ((1 - \alpha)y + \alpha x)\|^2 = \max_{x \in \Delta_c} \frac{\alpha}{2} \|y - x\|^2 = \frac{\alpha}{2} \max_{i \in [c]} \|y - \mathbb{1}_i\|^2 \, .$$

We rewrite the problem

$$\frac{\alpha}{2} \max_{i \in [c]} \|y - \mathbb{1}_i\|^2 = \frac{\alpha}{2} \max_{i \in [c]} \left\{ \sum_{j \neq i} y_j^2 + (y_i - 1)^2 \right\}$$

Note that for $a \in [0, 1]$, $a^2$ is monotonically increasing and $(1 - a)^2$ is monotonically decreasing. Hence the first summand of the right hand side is maximized by choosing the largest entries of $y$ and second summand of the right hand side is maximized by choosing the smallest entry of $y$. We conclude that $\arg\max_{i \in [c]} y_i$ is an optimizer of Equation (2).

**Case 2.** Note that $v \mapsto \text{CEL}(w, v)$ is continuous for $v \in \text{Ext}(\Delta_c)$ and $w \in \Delta_c$ and convex for $v \in \Delta_c$. By assumption, $y \in \text{Int}(\Delta_c)$ and $\alpha > 0$, hence $\alpha x + (1 - \alpha)y \in \text{Int}(\Delta_c)$, meaning that $x \mapsto \text{CEL}(y, \alpha x + (1 - \alpha)y)$ is continuous for all $x \in \Delta_c$. Hence by Lemma 4, there is an optimizer of Equation (2) on an extreme point of $\Delta_c$, i.e., the one-hot vector $\mathbb{1}_i$ for all $i \in [c]$. We explicit Equation (2),

$$\max_{x \in \Delta_c} \text{CEL}(x, (1 - \alpha)y + \alpha y) = \max_{x \in \Delta_c} -\sum_{i=1}^{c} x_i \log((1 - \alpha)y_j + \alpha x_i).$$

Since Equation (2) admits an optimizer on an extreme point of $\Delta_c$, we can write

$$k = \arg\max_{j \in [c]} \left\{ -y_j \log((1 - \alpha)y_j + \alpha) - \sum_{i \neq j}^{c} y_i \log((1 - \alpha)y_i) \right\}.$$

We rewrite the problem as follows

$$\max_{j \in [c]} - \sum_{i=1}^{c} y_i \log((1-\alpha)y_i) + y_j \log((1-\alpha)y_j) - y_j \log((1-\alpha)y_j + \alpha)$$

$$= - \sum_{i=1}^{c} y_i \log((1-\alpha)y_i) + \max_{j \in [c]} y_j \log \left( \frac{(1-\alpha)y_j}{(1-\alpha)y_j + \alpha} \right)$$

Note that $\frac{(1-\alpha)y_j}{(1-\alpha)y_j + \alpha} < 1$ for all $\alpha \in [0,1]$ and $y_j \in (0,1)$, hence the logarithm is negative and since $y_j > 0$, the term we want to optimize is monotonically decreasing. We conclude that $k = \arg\min_{j \in [c]} y_j$ optimizes the problem and $\mathbb{1}_k$ is an optimizer of Equation (2). $\square$

**Corollary 7** (HIPS+LMA). *Consider the following optimization problem*

$$\max_{\bar{Y}_{\mathcal{B}}(x) \in \mathcal{A}} \mathcal{L}\left((1-\alpha)\bar{Y}_{\mathcal{H}}(x) + \alpha \bar{Y}_{\mathcal{B}}(x), \bar{Y}_{\mathcal{H}}(x)\right), \tag{4}$$

*where $\mathcal{A} \overset{\text{def}}{=} \text{conv}\{(Y_i^{t+1}(x))_{i \in \mathcal{H}}\}$. Then under the assumption of Lemma 5, the problem admits a solution on an extreme point of $\mathcal{A}$.*

*Proof of Corollary 7.* As shown in the proof of Lemma 5, for both cases $\mathcal{L}$ is differentiable and convex, hence the statement follows directly from Lemma 4. $\square$

### E.3 COUNTEREXAMPLE

Consider the following three predictions

$$y_1 = \begin{pmatrix} 0.7 \\ 0.2 \\ 0.1 \end{pmatrix}, \quad y_2 = \begin{pmatrix} 0.8 \\ 0.1 \\ 0.1 \end{pmatrix}, \quad y_3 = \begin{pmatrix} 0 \\ 0 \\ 1 \end{pmatrix}, \tag{5}$$

which all lie in $\Delta_3$. Then the coordinate-wise median (which is a special case of the 50 % coordinate-wise trimmed mean) would yield $\tilde{y} = (0.7, 0.1, 0.1)^T$, which lies outside of the simplex.

