# OpenReview forum: "On the Byzantine-Resilience of Distillation-Based Federated Learning"
_ICLR.cc/2025/Conference — ICLR 2025 Poster_

### Official Review · Reviewer_ktFr · 2024-10-27

**Soundness:** 3
**Presentation:** 2
**Contribution:** 2
**Rating:** 6
**Confidence:** 4

**Summary:**

This paper explores the robustness of FedDistill, a knowledge distillation federated learning (FL) method, under Byzantine settings, where some clients act maliciously to disrupt training. The paper introduces two new Byzantine attacks, Loss Maximization Attack (LMA) and Class Prior Attack (CPA), that exploit this prediction-sharing approach. It also presents ExpGuard, a novel defense mechanism to counteract these attacks. The authors finally propose the HIPS method to obfuscate attacks, making them harder to detect. Experimental results demonstrate that FedDistill is more resilient than traditional FL aggregation methods.

**Strengths:**

1. Considering KD-based FL in a Byzantine setting is a good idea.
2. Propose two effective attacks for KD-based FL and a defending method. Additionally, the authors introduce a method that obfuscates attacks to make them harder to detect.
3. The experimental results show the effectiveness of the proposed methods.

**Weaknesses:**

1. The paper only considers iid setting. Although the authors claim that non-iid setting in FL poses serious issues (line 122), they don't specify what kind of issue makes it unsuitable for KD-based FL. Moreover, non-iid setting is essential in FL and is closer to realistic scenarios.
2. The scale of experiments is limited. The paper only considers cross-silo settings (a small number of clients) and lacks cross-device settings (over a hundred clients).
3. The ExpGuard assumes that the Byzantine clients are identical during training, which may be not the case in reality.

**Questions:**

1. In line 328 and figure 2, shouldn't the least likely class be _dog_ rather than _cat_?
2. CPA attack requires a pretained model to obtain the similarity matrix, how do you get it in the experiments? Is this effective in a more realistic setting?
3. Will the distribution shift between clients' data and public data affect the performance of the server's model? It's better to provide additional experiments or theoretical analysis on how different levels of distribution shifts between client and public data might impact the server model's performance. This could help readers better understand the limitations and robustness of the proposed approach.

---

> ### Author Response · Authors · 2024-11-15
> **Reply to Reviewer ktFr**
>
> Thank you for your feedback, please let us address your concerns in detail.
>
> >The paper only considers iid setting. Although the authors claim that non-iid setting in FL poses serious issues (line 122), they don't specify what kind of issue makes it unsuitable for KD-based FL. Moreover, non-iid setting is essential in FL and is closer to realistic scenarios.
>
> We do not claim that non-iid data is specifically problematic for KD-based FL as opposed to FedAVG. While the Byzantine-resilient FedAVG literature is extensive, unfortunately, research on Byzantine-resilient FedDistill is currently limited. Although addressing the non-IID case is a valuable direction, it is significantly more complex to provide any meaningful advancements in the non-IID byzantine setting. Up to now, for FedDistill, only few works have discussed the IID byzantine setting, which we focus on. It is essential to first gain a clear understanding of the IID setting. This approach mirrors the development of the Byzantine-resilient FedAVG literature, where focusing initially on IID data established a foundation for later addressing non-IID challenges.
>
> >The scale of experiments is limited. The paper only considers cross-silo settings (a small number of clients) and lacks cross-device settings (over a hundred clients).
>
> KD-based FL methods struggle in cross-device settings with standard ML datasets. As the number of clients increases, the fixed dataset size means less private data per client, making initial model training challenging. This issue is particularly problematic before the first communication round. We have addressed this in Appendix C with ablation studies.
>
> >The ExpGuard assumes that the Byzantine clients are identical during training, which may be not the case in reality.
>
> This assumption is standard for defence methods that rely on historical information, such as (Alistarh et al.,2018; Allen-Zhu et al., 2021; Karimireddy et al., 2021). Without this assumption, one cannot use historical information. In fact Karimireddy et al., 2021 show that for FedAVG-type algorithms, convergence is impossible in this scenario, see their Remark 3.
>
> >In line 328 and figure 2, shouldn't the least likely class be dog rather than cat?
>
> Thanks for pointing out this mistake, we have fixed it.
>
> >CPA attack requires a pretained model to obtain the similarity matrix, how do you get it in the experiments? Is this effective in a more realistic setting?
>
> In the experiments, we used publicly available models pretrained on the datasets in question. We then ran inference on the public dataset and computed the covariance matrix between the class probabilities of those predictions. We will add a more detailed description to the appendix.
>
> While we have noted in the paper that having a pretrained model is not necessarily a realistic assumption (lines 347-350), our goal was to search for an attack that would be able to considerably perturb the training process. As one can see from the experimental results, even under the strong assumption that one has access to a good similarity matrix, the impact of CPA stays limited. A more realistic approach might be to use prior knowledge about class relationships or simply use the server model from the last round. But we expect these methods to work at best a good as using a pretrained model for obtaining the similarity matrix.
>
> >Will the distribution shift between clients' data and public data affect the performance of the server's model? It's better to provide additional experiments or theoretical analysis on how different levels of distribution shifts between client and public data might impact the server model's performance. This could help readers better understand the limitations and robustness of the proposed approach.
>
> Indeed, you are correct, a significant difference between clients' data distribution and the public dataset distribution will hurt the performance of KD-based FL. While KD-based FL certainly provides a lot of advantages over methods that share parameters, this is indeed an issue, when relying on a public dataset, as such algorithms do. However, KD-based FL is widely used and the goal of this paper is to understand the impact of byzantine clients in this setting. Please let us note, that we are not advocating KD-based FL as a better method than anything else. Our work is constrained to the byzantine analysis.
>
> We thank you for your feedback. We hope to have addressed all your concerns. please let us know if you have further comments that help us to improve the paper.

---

> > ### Author Response · Authors · 2024-11-21
> >
> > Dear Reviewer ktFr, thanks again for your review. We want to remind you that we are available to answer any remaining questions or concerns you may have. We believe our rebuttal addresses all the points you raised, please consider increasing your rating of the paper if you agree. We are happy to clarify any further issues until the end of the discussion period. Thank you for your time and consideration!

---

> > ### Comment · Reviewer_ktFr · 2024-11-22
> >
> > Thank you for the authors' response, which has addressed most of my concerns. However, I strongly recommend that the authors discuss the non-iid setting. While this may not yet be broadly explored in the field of FedDistill, it remains a common and realistic scenario in federated learning. Importantly, client heterogeneity can significantly impact the server's ability to distinguish Byzantine clients.

---

> > > ### Author Response · Authors · 2024-11-24
> > >
> > > Thank you for your response, we are glad to hear that we have addressed most of your concerns.
> > >
> > > > Thank you for the authors' response, which has addressed most of my concerns. However, I strongly recommend that the authors discuss the non-iid setting. While this may not yet be broadly explored in the field of FedDistill, it remains a common and realistic scenario in federated learning. Importantly, client heterogeneity can significantly impact the server's ability to distinguish Byzantine clients.
> > >
> > > Although we argued that it is essential to first gain a clear understanding of the IID setting, we fully agree with you that it is of high importance to discuss KD-based FL in the non-IID setting. To that end, we implemented the Dirichlet distribution with parameter alpha, which is commonly used to generate non-IID data in the FL literature [cf. e.g. Lin et al., 2020 as in our references]. Here, we used the same setting as in our paper, except that the client datasets do not follow a uniform class distribution, but rather a skewed one depending on the parameter alpha, where smaller alpha means more heterogeneity and vice-versa. Below we provide complete experiments for the CINIC-10 dataset, and will follow up with the results on CIFAR-10, which we will be able to provide within the discussion phase.
> > >
> > > We ran all attacks and defences varying alpha between 10, 5 and 1. The four tables below distinguish the different defence mechanisms (highlighted in parentheses), providing the final test accuracies given different attacks (given as the rows) and for the IID case as well as for the non-IID cases with different alphas. For the attacks without HIPS, we see that not only does E+F achieve the best resilience among all defences, but the difference is even stronger than in the IID setting, especially for small alpha values. In fact, for alpha=1, both GM and Cronus perform worse than the mean, while the accuracy E+F reaches is more than double compared to using the mean. As opposed to other defence methods considered here, E+F maintains its robustness for higher levels of heterogeneity, making it a suitable choice for non-IID byzantine-resilient aggregation.
> > >
> > >
> > > As in the IID case, we observe that the mean is slightly more robust towards HIPS attacks than E+F. However, the mean is not resilient against attacks without HIPS. Therefore, if the server does not know what kind of attacks to expect, it is still preferable to choose E+F as it will maintain a good worst-case performance at the cost of a slight decrease in performance against HIPS attacks. We see that HIPS is in general much more effective in the non-IID case than in the IID case. This is intuitive as increasing the heterogeneity of the per-client-data leads to more heterogeneous predictions. Recall that HIPS reduces the attack space from the whole probability simplex to the convex hull of the honest clients predictions. Therefore, higher honest prediction heterogeneity leaves more space for the attacker to not be detected while disturbing the server.
> > >
> > > ### **CINIC-10 attacks (mean)**
> > > |          | iid      | alpha=10 | alpha=5  | alpha=1  |
> > > |----------|----------|----------|----------|----------|
> > > | RLF      | 76.9 | 75.5 | 73.9 | 62   |
> > > | CPA      | 45.9 | 37.2 | 28.1 | 18.5 |
> > > | LMA      | 54.6 | 52.0 | 47.4 | 27.2 |
> > > | HIPS+CPA | 74.2 | 60.7 | 53.4 | 32.6 |
> > > | HIPS+LMA | 75.3 | 61.2 | 52.8 | 28.9 |
> > >
> > > ### **CINIC-10 attacks (E+F)**
> > > |          | iid      | alpha=10 | alpha=5 | alpha=1 |
> > > |----------|----------|----------|---------|---------|
> > > | RLF      | 78.8 | 75.8     | 73.0    | 63.8    |
> > > | CPA      | 73.3 | 77.9     | 73.6    | 58.4    |
> > > | LMA      | 77.4 | 75.8     | 73.8    | 62.0    |
> > > | HIPS+CPA | 72.9 | 57.4     | 41.4    | 24.3    |
> > > | HIPS+LMA | 73.39 | 59.1     | 44.0    | 24.4    |
> > >
> > > ### **CINIC-10 attacks (GM)**
> > > |          | iid      | alpha=10 | alpha=5  | alpha=1  |
> > > |----------|----------|----------|----------|----------|
> > > | RLF      | 79.3 | 76.3 | 75.2 | 62.5 |
> > > | CPA      | 71.2 | 44.5 | 37.6 | 9.5  |
> > > | LMA      | 75.0 | 55.3 | 47.2 | 10.4 |
> > > | HIPS+CPA | 65.8 | 38.6 | 31.4 | 18.4 |
> > > | HIPS+LMA | 68.7 | 36.9 | 30.1 | 15.1 |
> > >
> > >
> > > ### **CINIC-10 attacks (Cronus)**
> > > |          | iid      | alpha=10 | alpha=5  | alpha=1  |
> > > |----------|----------|----------|----------|----------|
> > > | RLF      | 76.6 | 70.8 | 62.6     | 39.1 |
> > > | CPA      | 65.9 | 40.1 | 25.7 | 9.1  |
> > > | LMA      | 71.1 | 51.4 | 35.1 | 10.0     |
> > > | HIPS+CPA | 66.4 | 38.3 | 29.5 | 17.4 |
> > > | HIPS+LMA | 67.7 | 37.3 | 28.6 | 16.1 |
> > >
> > > We hope that this addresses your concern. We thank you again for your remark and will add a detailed discussion of the non-IID setting to the final manuscript.

---

> > > > ### Comment · Reviewer_ktFr · 2024-11-25
> > > >
> > > > Thanks for the detailed reply and it has addressed all my concerns. The results of the non-iid setting look interesting. I increase my score to 6.

---

> > > > > ### Author Response · Authors · 2024-11-25
> > > > >
> > > > > Thank you for improving your score. We're pleased to know that your concerns have been resolved.
> > > > >
> > > > > As promised, we provide the experimental results for CIFAR-10. The results follow the same pattern as those for CINIC-10 layed out above. This further reinforces the validity of our contributions in the non-iid setting.
> > > > >
> > > > > ### **CIFAR-10 attacks (mean)**
> > > > >
> > > > > |          | iid  | alpha=10 | alpha=5 | alpha=1 |
> > > > > |----------|------|----------|---------|---------|
> > > > > | RLF      | 84.6 | 83.9     | 80.6    | 71.0    |
> > > > > | CPA      | 68.4 | 57.9     | 38.2    | 15.4    |
> > > > > | LMA      | 73.4 | 69.5     | 65.6    | 35.5    |
> > > > > | HIPS+CPA | 85.0 | 76.2     | 59.5    | 32.3    |
> > > > > | HIPS+LMA | 84.9 | 75.6     | 54.0    | 32.0    |
> > > > >
> > > > >
> > > > > ### **CIFAR-10 attacks (E+F)**
> > > > >
> > > > > |          | iid  | alpha=10 | alpha=5 | alpha=1 |
> > > > > |----------|------|----------|---------|---------|
> > > > > | RLF      | 85.4 | 83.8     | 80.3    | 69.1    |
> > > > > | CPA      | 85.5 | 85.6     | 83.5    | 52.9    |
> > > > > | LMA      | 85.4 | 85.0     | 83.4    | 74.0    |
> > > > > | HIPS+CPA | 57.7 | 68.3     | 42.8    | 25.3    |
> > > > > | HIPS+LMA | 68.5 | 68.2     | 44.6    | 23.0    |
> > > > >
> > > > >
> > > > > ### **CIFAR-10 attacks (GM)**
> > > > >
> > > > > |          | iid  | alpha=10 | alpha=5 | alpha=1 |
> > > > > |----------|------|----------|---------|---------|
> > > > > | RLF      | 85.4 | 73.1     | 82.4    | 85.2    |
> > > > > | CPA      | 78.4 | 67.4     | 43.8    | 10.4    |
> > > > > | LMA      | 83.3 | 75.0     | 58.7    | 9.6     |
> > > > > | HIPS+CPA | 55.2 | 59.7     | 30.2    | 13.5    |
> > > > > | HIPS+LMA | 58.4 | 45.6     | 28.2    | 10.0    |
> > > > >
> > > > > ### **CIFAR-10 attacks (Cronus)**
> > > > >
> > > > > |          | iid  | alpha=10 | alpha=5 | alpha=1 |
> > > > > |----------|------|----------|---------|---------|
> > > > > | RLF      | 84.7 | 79.4     | 75.3    | 42.7    |
> > > > > | CPA      | 74.5 | 60.0     | 34.9    | 10.1    |
> > > > > | LMA      | 80.6 | 71.8     | 48.0    | 7.8     |
> > > > > | HIPS+CPA | 54.8 | 48.1     | 28.9    | 16.8    |
> > > > > | HIPS+LMA | 43.9 | 42.7     | 21.0    | 10.4    |

---

### Official Review · Reviewer_Zj4j · 2024-11-02

**Soundness:** 3
**Presentation:** 3
**Contribution:** 3
**Rating:** 6
**Confidence:** 4

**Summary:**

This paper studies the Byzantine resilience of federated knowledge distillation under IID data distributions. The authors start by analyzing the vanilla FedDistill and illustrate the vulnerabilities of FedAvg in the presence of Byzantine adversaries. The authors further propose three additional Byzantine attacks tailored for distillation-based federated learning methods. In response to enhance the federated distillation-based methods, the authors propose ExpGuard.

**Strengths:**

* The paper is, in general, well written.
* The proposed algorithms and the attack strategies are intuitive and easy to understand.

**Weaknesses:**

* The most obvious weakness of the paper is that it addresses the IID data distributions. In contrast, the current state-of-the-art (SOTA) Byzantine resilient federated learning methods focus on non-IID data setups.
* In Section 4, the motivating example to compare the vanilla FedAvg algorithm and the FedDistill algorithm is rather unfair. This is because the adversaries for the vanilla FedAvg employ model poisoning strategies, while the ones in FedDistill use data poisoning.
* In Section 4, the presentation in **the byzantine resilience of FedDistill** can be improved. All the theoretical theorems and lemmas have been deferred to the Appendix, making it hard to appreciate the real significance of resilience.
* In Section 5, the loss minimization attack (LMA) is a bit contrary to the prior Byzantine federated learning literature, e.g., [Karimireddy et al. (2022), Allouah et al. (2023)], where they assume the server is curious but honest. Here, the server is not reliable in the sense that it can manipulate the generated model parameters to mislead the training. At one extreme, the server can completely replace the model parameters to destroy the whole training process.
* In Section 5.2, the authors did not explain why ExpGuard works, except by citing [Littlestone & Warmuth, 1994].
* It is recommended to group HIPS together with LMA & CPA and present the defending strategy after the attacking strategies.
* The experiments are not thorough enough in two regards. First, the authors need to compare with the performance with FedAvg under current SOTA Byzantine resilient algorithms, e.g., bucketing [Karimireddy et al. (2022)] and [Allouah et al. (2023)] NNM. Second, the ablation experiments do not consider varying the number of Byzantine clients. The algorithms may struggle under benign settings.

**Questions:**

* In Section 4, is it possible to compare the vanilla FedAvg and FedDistill under the same Byzantine adversaries?
* In Section 4 **the byzantine resilience of FedDistill**, it seems that the authors consider the full-batch gradient only. However, stochastic gradients are the ones that the machine learning community uses in practice. How will the choice of stochastic gradients affect the problem?
* In Section 4, the authors claim that previous work on byzantine FedDistill focuses mostly on basic label-flipping attacks. Is it possible to consider another classic data poisoning attack: data injection?
* In Section 5, is the loss minimization attack a plug-in attack on top of the original data poisoning attack at each Byzantine client?
* In Section 5, can the authors further justify ExpGuard? In particular, why does the exponential weight work, but the other choices won't?
* In Section 6, can authors conduct more thorough experiments to compare (1) FedDistill algorithm with FedAvg under SOTA defense (2) the varying number of Byzantine clients?

---

> ### Author Response · Authors · 2024-11-15
> **Reply 1/3 to Reviewer Zj4j**
>
> We thank you for your detailed review. We will address your concerns in detail, hopefully resolving some misunderstandings.
>
> ### Weaknesses
> >The most obvious weakness of the paper is that it addresses the IID data distributions. In contrast, the current state-of-the-art (SOTA) Byzantine resilient federated learning methods focus on non-IID data setups.
>
> While the Byzantine-resilient FedAVG literature is extensive, unfortunately, research on Byzantine-resilient FedDistill is currently limited. Although addressing the non-IID case is a valuable direction, it is significantly more complex to provide any meaningful advancements in the non-IID byzantine setting. Up to now, for FedDistill, only few works have discussed the IID byzantine setting, which we focus on. It is essential to first gain a clear understanding of the IID setting. This approach mirrors the development of the Byzantine-resilient FedAVG literature, where focusing initially on IID data established a foundation for later addressing non-IID challenges.
>
> >In Section 4, the motivating example to compare the vanilla FedAvg algorithm and the FedDistill algorithm is rather unfair. This is because the adversaries for the vanilla FedAvg employ model poisoning strategies, while the ones in FedDistill use data poisoning.
>
> We note that byzantine clients *cannot* poison the model in FedDistill, as they can only influence the server model by modifying the labels on the public dataset, i.e., data poisoning. So the unfairness you pointed out is precisely what makes FedDistill so resilient. We refer to Lines 219-244 for a more detailed discussion of how the threat vectors of byzantine clients differ between FedAVG and FedDistill.
>
> >In Section 4, the presentation in the byzantine resilience of FedDistill can be improved. All the theoretical theorems and lemmas have been deferred to the Appendix, making it hard to appreciate the real significance of resilience.
>
> Thank you for your suggestion. We will highlight our theoretical results more prominently in the final revision of the paper.
>
> >In Section 5, the loss minimization attack (LMA) is a bit contrary to the prior Byzantine federated learning literature, e.g., [Karimireddy et al. (2022), Allouah et al. (2023)], where they assume the server is curious but honest. Here, the server is not reliable in the sense that it can manipulate the generated model parameters to mislead the training. At one extreme, the server can completely replace the model parameters to destroy the whole training process.
>
> We are not entirely sure whether we correctly understand your remark. LMA is an attack that byzantine clients perform, to find adversarial predictions, which are sent to the server in order to negatively influence the training process. The challenge for the server is to reduce the impact of byzantine predictions, to eventually learn a good model in spite of the byzantine clients. Could you elaborate how this is related to the server being "curious but honest"?
>
> >In Section 5.2, the authors did not explain why ExpGuard works, except by citing [Littlestone & Warmuth, 1994]. [...] In Section 5, can the authors further justify ExpGuard? In particular, why does the exponential weight work, but the other choices won't?
>
> The goal of the server is to discard the byzantine predictions and to only use the predictions of the honest clients. In order to have an impact on the aggregated predictions, the byzantine clients must consistently make predictions that differ from those of the honest clients. Therefore we expect that the predictions made by the byzantine clients are on average further away from the mean honest prediction, which the server does not know. However, we know that robust aggregation methods such as the GM reduce the impact of byzantines and are hence close to the mean honest prediction. Hence, ExpGuard takes this robust aggregator as a proxy for the honest mean predictions and tracks how much each client differs from it on average. Specifically, this difference is quantified by the *outlier score*, i.e., the sum of distances from the robust aggregator over all samples. We expect clients that have a higher outlier score, i.e., that are on average further away from the proxy, to be more likely to be byzantine and vice-versa. ExpGuard assigns a weight to each client based on this outlier score via exponential weights update.
>
> We thank you for pointing out this gap in our explanation, we will add it.
>
> >It is recommended to group HIPS together with LMA & CPA and present the defending strategy after the attacking strategies.
>
> Thank you, we agree that this could be improved, we will change the order of presentation in the final version of this paper.

---

> > ### Author Response · Authors · 2024-11-15
> > **Reply 2/3 to Reviewer Zj4j**
> >
> > >The experiments are not thorough enough in two regards. First, the authors need to compare with the performance with FedAvg under current SOTA Byzantine resilient algorithms, e.g., bucketing [Karimireddy et al. (2022)] and [Allouah et al. (2023)] NNM. Second, the ablation experiments do not consider varying the number of Byzantine clients. The algorithms may struggle under benign settings. [...] In Section 6, can authors conduct more thorough experiments to compare (1) FedDistill algorithm with FedAvg under SOTA defense (2) the varying number of Byzantine clients?
> >
> > Let us address each of the two points separately.
> > (1) The goal of this article is to examine the byzantine-resilience of KD-based FL algorithms as these methods are becoming more popular, not to show that either FedAVG or FedDistill are more byzantine-resilient. Both methods have their advantages and drawbacks.
> >
> > We do not claim that FedDistill is in general more byzantine-resilient than FedAVG. We merely pointed out that vanilla FedDistill is remarkably byzantine-resilient as opposed to vanilla FedAVG, which can be perturbed arbitrarily by a single byzantine client. We used this as a motivating example.
> >
> > While we think that a direct comparison is hard to interpret, we nevertheless performed the experiments you suggested in order to address your concern. We implemented D-SGD with momentum and using nearest neighbour mixing (NNM) with coordinate-wise median (CWMED) aggregation for CINIC-10 against the ALIE attack as layed out in [Allouah et al. (2023)] with 20 total clients. All other hyperparameters are as described in the paper. Without attacks, D-SGD reaches an accuracy of 80.8%±1.5 when using the mean aggregation and 77.7%±1.6 using NNM+CWMED. The table below lists the accuracy for different attack magnitudes $\eta$ and number of byzantine clients.
> >
> > |\# byz. clients \| $\eta$| 0.05         | 0.1         | 0.5             | 1               | 2            |
> > |------------------------|--------------|-------------|-----------------|-----------------|--------------|
> > | 1                      | 78.7% ± 0.4  | 78.3% ± 0.7 | **69.0%** ± 2.4 | 75.8% ± 0.0     | 79.9% ± 0.0  |
> > | 5                      | 77.2% ± 0.8  | 70.0% ± 4.0 | 27.6% ± 1.5     | **24.1%** ± 0.2 | 71.5% ± 0.0  |
> > | 9                      | 55.8% ± 26.0 | 55.3% ± 5.8 | 25.8% ± 2.9     | **19.8%** ± 0.2 | 62.2% ± 13.7 |
> >
> > Byzantine clients can still almost completely hinder the learning process, decreasing the accuracy to under 20% with 9 byzantine clients. This is a reduction in accuracy by a factor of more than 4 compared to the setting without byzantine clients as opposed to a factor 1.1 in FedDistill. Note that we chose a fixed attack amplitude for ALIE without further optimizing it to inflict even more damage, as proposed in [Allouah et al. (2023)]. We hope that these results addressed your concern regarding the performance of our methods.
> >
> > (2) We always consider the maximal number of byzantine clients such that $\alpha<0.5$. We noticed, that in many experiments involving FedDistill, fewer byzantine clients had almost no measurable impact.
> >
> > *Performance in the benign setting:*
> > We are currently running the experiments to provide the baseline performance of each defence in the benign setting and will post them as soon as possible. In the mean time, please note that we always provide the baseline performance of FedDistill with mean aggregation, denoted as BA, and that the performance of our method EG+F is not far off from that baseline, even in the presence of byzantine attacks.
> >
> > *Ablations for the number of clients:*
> > We ran EG+F on CINIC-10 varying the number of byzantine clients. We chose LMA with and without HIPS as attack, as this was the strongest attack on CINIC-10 for 9 byzantine clients. The baseline performance of FedDistill using mean aggregation without byzantine clients is 80.2%±0.1. We will provide full ablations in the final version of this paper.
> >
> > |          | 0         | 3         | 5         | 7         | 9         |
> > |----------|-----------|-----------|-----------|-----------|-----------|
> > | HIPS+LMA | 80.5%±1.2 | 79.7%±1.0 | 79.8%±0.4 | 78.9%±0.1 | 72.9%±0.7 |
> > | LMA      | 80.5%±1.2 | 79.7%±0.1 | 79.9%±0.0 | 79.9%±0.2 | 77.4%±1.1 |

---

> > > ### Author Response · Authors · 2024-11-15
> > > **Reply 3/3 to Reviewer Zj4j**
> > >
> > > ### Questions
> > >
> > > > In Section 4, is it possible to compare the vanilla FedAvg and FedDistill under the same Byzantine adversaries?
> > >
> > > If by same byzantine adversaries you are referring to the same attacks, then no, because the attack space is different. In FedAVG, the byzantine clients operate over the parameter space, while in FedDistill, they operate over the probability simplex of predictions. For example, the gaussian distribution is defined over the real numbers, yet for FedDistill, byzantine clients are constrained to the probability simplex. One could consider some other distribution defined over the simplex such as the uniform or the Dirichlet distribution, but the random label flip will almost surely be stronger as the predictions always lie on the extreme points of the probability simplex, maximizing the amplitude of the attack.
> > >
> > > >In Section 4 the byzantine resilience of FedDistill, it seems that the authors consider the full-batch gradient only. However, stochastic gradients are the ones that the machine learning community uses in practice. How will the choice of stochastic gradients affect the problem?
> > >
> > > Thank you for pointing this out. The analysis can easily be extended to stochastic gradients. We will add it in the final version of the paper.
> > >
> > > >In Section 4, the authors claim that previous work on byzantine FedDistill focuses mostly on basic label-flipping attacks. Is it possible to consider another classic data poisoning attack: data injection?
> > >
> > > In FedDistill, the only way the clients can impact the server is via their predictions on the public dataset. We assume that they cannot modify the public dataset itself. Since the dataset has to be publicly available it seems unrealistic for a byzantine client to be able to modify it. Is that what you refer to with "data injection", adding new samples?
> > >
> > > >In Section 5, is the loss minimization attack a plug-in attack on top of the original data poisoning attack at each Byzantine client?
> > >
> > > No, all attacks we consider consist of choosing the byzantine clients predictions in a certain way. As such, one has to decide which attack to choose. We are not sure what you mean by the "original data poisoning attack".
> > >
> > > We thank you for your feedback. We hope to have addressed all your concerns. please let us know if you have further comments that help us to improve the paper.

---

> > > ### Author Response · Authors · 2024-11-21
> > >
> > > Here are the experimental results we promised regarding the performance of our defenses in benign settings. We compared the performance of the defenses against the mean baseline under these conditions. As shown in the table, the accuracy remains comparable, with only minimal drops in accuracy. These comparisons were conducted without modifying any other hyperparameters, suggesting that these small variations could likely be mitigated by adjusting hyperparameters.
> > >
> > >
> > > |                     | CIFAR-10 | CINIC-10 | CIFAR-100 | Clothing-1M |
> > > |---------------------|----------|----------|-----------|-------------|
> > > | FedDistill (E+F)    | 86.2±1.0 | 80.5±0.2 | 64.9±1.9  | 68.0±0.2    |
> > > | FedDistill (GM)     | 87.0±0.1 | 81.1±0.5 | 66.4±0.4  | 69.4±0.0    |
> > > | FedDistill (Cronus) | 86.4±0.3 | 80.4±0.2 | 62.6±0.8  | 64.9±0.2    |
> > > | FedDistill (mean)   | 87.7±1.2 | 80.2±0.1 | 66.8±0.5  | 69.0±0.3    |

---

> > > > ### Comment · Reviewer_Zj4j · 2024-11-21
> > > >
> > > > I want to thank the authors for their detailed and constructive responses. I think they have addressed my concerns. Therefore, I decided to increase my score to 6.

---

> > ### Comment · Reviewer_Zj4j · 2024-11-16
> >
> > Thank you, the authors, for their responses to my comments. They addressed most of my concerns in this part, but I have two presentation suggestions for the authors.
> >
> > * **Model poisoning.**
> >
> > While I agree with the authors that the outcomes of distillation-based methods are reflected solely by the predictions, that does not mean Byzantine clients cannot poison the model. This is because the poisoned data can result from the polluted model. The equivalent results do not imply impossibility. I suggest the authors revise their presentation in a future revision to reflect such equivalence and so to highlight their focus on data poisoning.
> >
> > * **LMA attack.**
> >
> > My initial evaluations of the LMA attack scheme were based on the impression that only the server has access to the mean of honest client predictions and Byzantine client predictions, which is based on the statement (line 320) "$\mathcal{L}$ that the server is trying to minimize". After the authors' responses, I realized that since the Byzantine clients here are omnicious, they also have access to such sensitive information. I suggest the authors add a clarification after that statement to emphasize that Byzantine clients are also able to access the loss.

---

> > > ### Author Response · Authors · 2024-11-17
> > >
> > > Thank you for your fast response! We are happy to hear to have addressed your concerns.
> > >
> > > > While I agree with the authors that the outcomes of distillation-based methods are reflected solely by the predictions, that does not mean Byzantine clients cannot poison the model. This is because the poisoned data can result from the polluted model. The equivalent results do not imply impossibility. I suggest the authors revise their presentation in a future revision to reflect such equivalence and so to highlight their focus on data poisoning.
> > >
> > > We suspect that there might be a misunderstanding here due to a mismatch in the terminology. Let us lay out our understanding of these terms in detail. We do not argue that the "outcomes of distillation-based methods are reflected solely by the predictions" but rather that the influence of the (byzantine) clients on distillation-based methods is restricted to their predictions on the public dataset. Further, it is not the poisoned data that results from the polluted model, but the inverse, so the poisoned data, i.e., the poisoned predictions on the public dataset, result in a polluted model. The impossibility of model poisoning comes by the definition of our setup.
> > >
> > > In your review, you initially mentioned the fact that we compared *model poisoning* attacks in FedAVG with *data poisoning* attacks in FedDistill. As we understand it, *model poisoning* refers to the direct modification of the model parameters, whereas *data poisoning* refers to modifying the data, which the model is trained on, to influence the model. Using this definition, the typical FedAVG attacks (e.g., Gaussian noise attack, among others) are model poisoning attacks. On the other hand, all attacks in FedDistill are data poisoning attacks, as the byzantine clients can only impact the server model indirectly via their predictions as the server uses the clients aggregated predictions as its training data. Note that here not the actual input data is poisoned, but rather the labels or predictions.
> > >
> > > We hope that this resolves the issue. We are happy to clarify this point in the paper, if needed.
> > >
> > > > My initial evaluations of the LMA attack scheme were based on the impression that only the server has access to the mean of honest client predictions and Byzantine client predictions, which is based on the statement (line 320) " that the server is trying to minimize". After the authors' responses, I realized that since the Byzantine clients here are omnicious, they also have access to such sensitive information. I suggest the authors add a clarification after that statement to emphasize that Byzantine clients are also able to access the loss.
> > >
> > > Please note that in the Preliminaries (specifically lines 156-158), we state that the byzantine clients have access to the predictions of all other clients. We will restate this when introducing the attacks, to avoid any confusion. Thank you for this remark. When we say that the clients have access to the loss (as you mentioned, line 320), we only mean that they have the knowledge which loss function the server uses, for example as in cross entropy or MSE, not anything data-related. We agree that this phrasing is ambiguous and will change it. Thank you!
> > >
> > > Please, let us further clarify the setting: The server receives the individual predictions of each client, but is unaware which clients are byzantine (Line 154,155). The byzantine clients can also access the individual predictions of the honest clients. In the case of LMA, we further assume that they are aware which loss function the server uses to train its model.
> > >
> > > We hope that this resolves all issues, if not, please let us know! If we understand you correctly, we have addressed your concerns, and we hope that you consider increasing your rating of the paper.

---

### Official Review · Reviewer_zFM4 · 2024-11-04

**Soundness:** 3
**Presentation:** 2
**Contribution:** 3
**Rating:** 6
**Confidence:** 4

**Summary:**

This paper evaluates the robustness of distillation based distributed learning to arbitrary (Byzantine) faults in the i.i.d setting.

The work illustrates the resilience of KD-based algorithms and analyzes how
byzantine clients can influence them. It also proposes two byzantine attacks and assess their ability to break existing
byzantine-resilient methods. It also proposes a method to further improve the byzantine resilience of KD-based FL algorithms.

**Strengths:**

the paper addresses an important and previously undressed question: the robustness of distributed learning in the distillation setting

**Weaknesses:**

the formal guarantees (left to appendix D) deserve a more prominent place in the paper, given that the core contribution of such a paper is to formally derive security guarantees and their limits when (in this case) the space of all possible attacks cannot rely on empirical validation

**Questions:**

What would be the effect of most comon robust aggregation schemes on KD-based FL? how would that change the bound in appendix D.1?

---

> ### Author Response · Authors · 2024-11-15
> **Reply to Reviewer zFM4**
>
> We thank you for your remark. Let us address your concerns in detail.
>
> > the formal guarantees (left to appendix D) deserve a more prominent place in the paper, given that the core contribution of such a paper is to formally derive security guarantees and their limits when (in this case) the space of all possible attacks cannot rely on empirical validation
>
> Thanks for that remark, we appreciate that you think our theoretical results should be highlighted more prominently. We agree. However, at submission, we thought that the readership of ICLR might potentially be more interested in empirical results. Yet, we have provided these theoretical results in the appendix, so if we understand correctly, your concern is mainly a matter of presentation? We will make the theoretical results more prominent in the final revision of the paper.
>
> > What would be the effect of most comon robust aggregation schemes on KD-based FL? how would that change the bound in appendix D.1?
>
> Most commonly used robust aggregation methods such as the coordinate-wise trimmed mean or the coordinate-wise median cannot be used to aggregate predictions, as the aggregated vectors do not necessarily lie in the probability simplex, see Appendix D.3 for a simple counter-example. This however is a necessary condition for a prediction vector to be meaningful.
>
> For aggregation methods that produce vectors within the probability simplex and have guarantees independent of statistical assumptions, we can tighten the error bound. For instance, using Allouah et al. (2023), we can bound the difference between the honest mean and the GM.
>
> We thank you again for your feedback, please let us know if you have further comments that help us to improve the paper.

---

> > ### Author Response · Authors · 2024-11-21
> >
> > Dear Reviewer zFM4, thanks again for your review. We want to remind you that we are available to answer any remaining questions or concerns you may have. We believe our rebuttal addresses all the points you raised, please consider increasing your rating of the paper if you agree. We are happy to clarify any further issues until the end of the discussion period. Thank you for your time and consideration!

---

> > > ### Comment · Reviewer_zFM4 · 2024-11-27
> > > **Increased score**
> > >
> > > Thanks to the authors for their reply, my score has been increased to reflect your work on updating the paper with our feedbacks.

---

### Official Review · Reviewer_4qUt · 2024-11-04

**Soundness:** 3
**Presentation:** 3
**Contribution:** 2
**Rating:** 6
**Confidence:** 5

**Summary:**

This paper first shows that distillation based method is more robust to conventional federated average using the Gaussian attack as an example. Then based on the observations, two labelling attack schemes are developed to break current distillation based robust algorithms and accordingly a new algorithm is developed to against the new attacks.

**Strengths:**

The paper is well written and constructed in a fluent structure. Extensive experiments have been conducted to verify the proposed ideas and methods. The proposed method is also somewhat new in the Knowledge distillation domain.

**Weaknesses:**

1. The comparison of FedAVG and KD-FL is a little weak and the experimental part is limited to KD-based methods, making the whole idea less convincing.
2. The proposed ExpGuard+F method is similar as the FLTrust methods, which is okay, however, the strength of the proposed method is not exactly reflected from the experiments. For example, is ExpGuard+F better than ExpGuard+GM and if yes, why?
3. The study of HIPS conveys the message that some attacks are hard to detect and combat, but we can use simple mean aggregation instead. I could not see the significance of this point right now.

**Questions:**

1. It is not clear what is the strength (mean/variance) of the Gaussian attack that is used to show the superiority of KD-FL against FedAvg.  Meanwhile, it is not fair to draw the conclusion that KD-based method is more resilient to FedAvg by one comparison between random Gaussian attack to FedAvg and the random label flipping for KD-FL. A fairer comparison could include some other attacks.
2. The study of HIPS is interesting, however, the result simply suggests that sometimes robust aggregation is worse than mean aggregation. To make this point more valuable, it might be better to have a trade-off analysis or provide some guidelines to how to choose robust aggregation methods.
3. The prediction for honest agent 4 in Figure 2 should be [0.61, 0.08, 0.31] to ensure the correct average prediction.
4. At least GM has been widely used in FedAVG for robust aggregation, therefore in line 387, when the authors say “As a result, applying them to aggregated parameters in Byzantine FedAVG is often impractical” does not make sense to me. It would be great if the authors either provide evidence supporting their claim about impracticality, or to revise their statement to better reflect the current state of practice in the field.
5. What is the baseline performance of FedAVG on the tested datasets? To me, KD-FL is more communication efficient as it only transfers predictions rather than model parameters, but is its performance better than or equivalent to FedAVG? If yes, then it makes sense to study KD-FL in Byzantine settings, otherwise, we can trade some communication for performance improvement and develop some communication-efficient and robust FedAVG algorithms.
5. ExpGuard shares some similarities with the FLTrust method, which is not new. It would be better to clarify the strength and difference of ExpGuard against FLTrust [1].
[1] Cao, Xiaoyu, Minghong Fang, Jia Liu, and Neil Zhenqiang Gong. "Fltrust: Byzantine-robust federated learning via trust bootstrapping." arXiv preprint arXiv:2012.13995 (2020).

---

> ### Author Response · Authors · 2024-11-15
> **Reply 1/2 to Reviewer 4qUt**
>
> Thank you for your feedback. We will address your concerns in detail.
>
> > It is not clear what is the strength (mean/variance) of the Gaussian attack that is used to show the superiority of KD-FL against FedAvg.
>
> Thank you for pointing this out. The attack consists of sending $\mathcal{N}(0,1)$ noise instead of the updated weights. We will add this information.
>
> > Meanwhile, it is not fair to draw the conclusion that KD-based method is more resilient to FedAvg by one comparison between random Gaussian attack to FedAvg and the random label flipping for KD-FL. A fairer comparison could include some other attacks.
>
> Please let us clarify, that we do not claim that FedDistill is in general more byzantine-resilient than FedAVG. The goal of this article is to examine the byzantine-resilience of KD-based FL algorithms as these methods are becoming more popular, not to show that either FedAVG or FedDistill are more byzantine-resilient. Both methods have their advantages and drawbacks. Figure 1 is a motivating example showing that it is easy to find simple attacks that completely prevent the server from training a model in vanilla FedAVG, while in FedDistill such simple attacks have very little impact.
>
>
> > The proposed ExpGuard+F method is similar as the FLTrust methods, which is okay, however, the strength of the proposed method is not exactly reflected from the experiments. For example, is ExpGuard+F better than ExpGuard+GM and if yes, why? [...] ExpGuard shares some similarities with the FLTrust method, which is not new. It would be better to clarify the strength and difference of ExpGuard against FLTrust [1].
>
> Thanks for pointing out this article, we were not aware of it and will add it to the related work. FLTrust differs from FedDistill in two key aspects. First, the approach of FLTrust is specific to FedAVG-type methods meaning that the analysis and intuition does not carry over to FedDistill. Second, while ExpGuard uses only the predictions of the clients on other samples and from previous rounds, i.e., information that is available anyways, FLTrust relies on an additional *labeled* dataset available only to the server in order to compute a trust score. This is a much stronger assumption.
>
> We discussed the performance of combining ExpGuard with different robust aggregators in lines 402-407 and provide a experimental results in Table 5, Appendix C. There, we see that ExpGuard+F significantly outperforms ExpGuard+GM, among others.
>
> >The study of HIPS is interesting, however, the result simply suggests that sometimes robust aggregation is worse than mean aggregation. To make this point more valuable, it might be better to have a trade-off analysis or provide some guidelines to how to choose robust aggregation methods.
>
> The point we want to make is that while the resilience of EG+F might be slightly worse than that of the mean against HIPS attacks, mean aggregation is not resilient at all against attacks without HIPS. Therefore, if the server does not know what kind of attacks to expect, it is still preferable to choose EG+F as it will maintain a good worst-case performance at the cost of a slight decrease in performance against HIPS attacks.
>
> >The prediction for honest agent 4 in Figure 2 should be [0.61, 0.08, 0.31] to ensure the correct average prediction.
>
> Thank you for pointing out this mistake, we will fix it.
>
> >At least GM has been widely used in FedAVG for robust aggregation, therefore in line 387, when the authors say “As a result, applying them to aggregated parameters in Byzantine FedAVG is often impractical” does not make sense to me. It would be great if the authors either provide evidence supporting their claim about impracticality, or to revise their statement to better reflect the current state of practice in the field.
>
> Thank you for pointing out this inaccuray. What we intended to say was that these methods can lead to a considerable computational overhead when applied to the parameters of large models, whereas the computational cost of aggregating predictions with these methods is negligible. We have changed the writing accordingly.

---

> > ### Author Response · Authors · 2024-11-15
> > **Reply 2/2 to Reviewer 4qUt**
> >
> > >What is the baseline performance of FedAVG on the tested datasets? To me, KD-FL is more communication efficient as it only transfers predictions rather than model parameters, but is its performance better than or equivalent to FedAVG? If yes, then it makes sense to study KD-FL in Byzantine settings, otherwise, we can trade some communication for performance improvement and develop some communication-efficient and robust FedAVG algorithms.
> >
> > Our aim is not to advocate for the use of KD-FL over FedAVG, as both come with specific advantages and drawbacks. KD-FL methods have attracted a lot of interest in the FL community, which is why we believe that it is important to examine the byzantine-resilience of KD-based FL algorithms.
> >
> > Also, note that many works used KD-FL *in combination* with FedAVG so it is not necessarily a question of one method versus the other. For example, Lin et al., 2020 use KD to make FedAVG more robust to non-iid data. Our goal was to specifically analyze the new threat vectors arising in byzantine KD-FL, which would also apply to such hybrid methods.
> >
> > In the table below, we provide a baseline of FedAVG compared to FedDistill. For FedDistill, we used the setup explained in the paper, i.e., 20 clients and 10 communication rounds (see Appendix C for more details). For FedAVG we follow this setup as closely as possible, i.e., we keep the same architectures and number of clients. For CIFAR-10 and CINIC-10, we also kept the number of communications rounds the same. However, for CIFAR-100 and Clothing-1M, we used 100 communication rounds as we were not able to achieve competitive performance with fewer communication rounds.
> >
> > We note that for Clothing-1M, the performance of FedAVG is considerably worse even with a hundred communication rounds. We suspect that this is due to the fact that FedAVG cannot leverage the unlabeled public dataset. This reinforces our point that one should not try to compare these two methods as they work under different assumptions.
> >
> > |            | CIFAR-10 | CINIC-10 | CIFAR-100 | Clothing-1M |
> > |------------|----------|----------|-----------|-------------|
> > | FedAVG     | 88.6±0.1 | 75.1±0.2 | 67.2±0.3  | 61.2±1.1    |
> > | FedDistill | 86.8±1.0 | 80.2±0.1 | 66.8±0.5  | 69.0±0.3    |
> >
> > We thank you for taking the time to review our manuscript, we appreciate it. We hope to have addressed all your concerns. Please let us know if you have further comments that help us to improve the paper.

---

> > > ### Author Response · Authors · 2024-11-21
> > >
> > > Dear Reviewer 4qUt,
> > > thank you again for your review. Regarding the experiments being limited to KD-based methods, which we addressed in our reply to your review, we wanted to let you know that we performed further experiments in that regard.
> > >
> > > While we still think that a direct comparison between FedAVG and KD-based methods is hard to interpret, we performed further experiments in order to address your concern. We implemented a SOTA defence [Allouah et al. (2023)]: Precisely, we use D-SGD with momentum and nearest neighbour mixing (NNM) with coordinate-wise median (CWMED) aggregation for CINIC-10 against the ALIE attack as layed out in [Allouah et al. (2023)] with 20 total clients. All other hyperparameters are as described in the paper. Without attacks, D-SGD reaches an accuracy of 80.8%±1.5 when using the mean aggregation and 77.7%±1.6 using NNM+CWMED. The table below lists the accuracy for different attack magnitudes $\eta$ and number of byzantine clients.
> > >
> > > |\# byz. clients \| $\eta$| 0.05         | 0.1         | 0.5             | 1               | 2            |
> > > |------------------------|--------------|-------------|-----------------|-----------------|--------------|
> > > | 1                      | 78.7% ± 0.4  | 78.3% ± 0.7 | **69.0%** ± 2.4 | 75.8% ± 0.0     | 79.9% ± 0.0  |
> > > | 5                      | 77.2% ± 0.8  | 70.0% ± 4.0 | 27.6% ± 1.5     | **24.1%** ± 0.2 | 71.5% ± 0.0  |
> > > | 9                      | 55.8% ± 26.0 | 55.3% ± 5.8 | 25.8% ± 2.9     | **19.8%** ± 0.2 | 62.2% ± 13.7 |
> > >
> > > Byzantine clients can still almost completely hinder the learning process, decreasing the accuracy to under 20% with 9 byzantine clients. This is a reduction in accuracy by a factor of more than 4 compared to the setting without byzantine clients as opposed to a factor 1.1 in FedDistill. Note that we chose a fixed attack amplitude for ALIE without further optimizing it to inflict even more damage, as proposed in [Allouah et al. (2023)]. We hope that these results addressed your concern regarding the performance of our methods.
> > >
> > > Please consider increasing your rating of the paper if we have addressed all of your concerns. If not, please let us know!

---

### Comment · Area_Chair_G5kW · 2024-11-26
**Response**

Dear Reviewers,

The authors have provided their rebuttal to your questions/comments. It will be very helpful if you can take a look at their responses and provide any further comments/updated review, if you have not already done so.

Thanks!

---

### Author Response · Authors · 2024-11-26
**Official comment regarding the latest revision**

As the discussion period ends, we again express our gratitude to all reviewers for their constructive feedback, assistance in improving our work, and increasing their scores towards acceptance. We thank the reviewers for highlighting that our paper addresses an important and previously unexplored question, commending the well-written structure, intuitive attack and defense strategies, the extensive experimental validation and noting the novelty and relevance of our contributions to the FL domain.

Let us quickly summarize the findings of the discussion period and the changes we have added to the updated PDF.

 - We have made our theoretical results more prominent by moving them to the main part of our manuscript.
 - We have extended the theoretical analysis from the deterministic to the stochastic setting.
 - We clarified the connection and comparison to FedAVG.
 - We have added the following ablations:
   - Baseline performance comparison of the presented defence methods in the benign setting.
   - Comparison between FedDistill and FedAVG in the benign setting).
   - Comparison between FedDistill and SOTA FedAVG byzantine-resilient methods, showing the performance benefits of FedDistill.
   - Performance of FedDistill in the byzantine setting when varying the number of byzantine clients.
- We have added experiments for the non-IID case, demonstrating that our results are consistent for both IID and non-IID client data.

We kindly ask the reviewers that have not responded to our rebuttal to let us know whether their concerns have been properly addressed, as your feedback is valuable to us and the discussion period is almost over. Thanks!

---

### Meta-Review · Area_Chair_G5kW · 2024-12-20

**Metareview:**

This paper studies use of knowledge distillation in federated learning to address the issues of heterogeneity and privacy. It is an interesting topic. While the reviewers were initially skeptical of this work, they all increased their scores as a result of the effort by authors in addressing their queries satisfactorily.

I recommend acceptance.

**Additional Comments On Reviewer Discussion:**

It has been a very fruitful discussion phase that will lead to an improved version of the paper.

---

### Decision · Program_Chairs · 2025-01-22

Accept (Poster)